



# Dynamical and hydrological changes in climate simulations of the last millennium

Pedro José Roldán-Gómez[1], Jesús Fidel González-Rouco[1], Camilo Melo-Aguilar[1], and Jason E. Smerdon[2]

[1]Instituto de Geociencias, Consejo Superior de Investigaciones Científicas - Universidad Complutense de Madrid, 28040 Madrid, Spain
[2]Lamont-Doherty Earth Observatory of Columbia University, Palisades, NY, United States of America

**Correspondence:** P. J. Roldán-Gómez (peroldan@ucm.es)

**Abstract.** Simulations of last millennium (LM) climate show that external forcing had a major contribution to the evolution of temperatures; warmer and colder periods like the Medieval Climate Anomaly (MCA; ca. 950-1250 CE) and the Little Ice Age (LIA; ca. 1450-1850 CE) were critically influenced by changes in solar and volcanic activity. Even if this influence is mainly observed in terms of temperatures, evidence from simulations and reconstructions show that other variables related to atmospheric dynamics and hydroclimate also were influenced by external forcing over some regions. In this work, simulations from the Coupled Model Intercomparison Project Phase 5 / Paleoclimate Modelling Intercomparison Project Phase 3 (CMIP5/PMIP3) are analyzed to explore the influence of external forcings on the dynamical and hydrological changes during the LM at different spatial and temporal scales. Principal Component (PC) analysis is used to obtain the modes of variability governing the global evolution of climate and to assess their correlation with the total external forcing at multidecadal to multicentennial timescales. For shorter timescales, a composite analysis is used to address the response to specific events of external forcing like volcanic eruptions. The results show coordinated long-term changes in global circulation patterns, which suggest expansions and contractions of the Hadley Cells and latitudinal displacements of Westerlies in response to external forcing. For hydroclimate, spatial patterns of drier and wetter conditions in areas influenced by the North Atlantic Oscillation (NAO), Northern Annular Mode (NAM) and Southern Annular Mode (SAM) and alterations in the intensity and distribution of monsoons and convergence zones are consistently found. Similarly, a clear short-term response is found in the years following volcanic eruptions. Although external forcing has a larger influence on temperatures, the results suggest that dynamical and hydrological variations over the LM exhibit a direct response to external forcing both at long and short timescales that is highly dependent on the particular simulation and model.

## 1 Introduction

Reconstructions and model simulations have shown that the evolution of temperatures during the last millennium (LM) was influenced both by external forcing and internal variability (Fernández-Donado et al., 2013; Schurer et al., 2013). This evolution was characterised by long-term changes, like the transition from the Medieval Climate Anomaly (MCA; ca. 950-1250 CE) to the Little Ice Age (LIA; ca. 1450-1850 CE) and during the period of anthropogenic warming. The MCA and the LIA



were periods of relatively warmer and cooler conditions, respectively (Diaz et al., 2011; Graham et al., 2010), which have been related to solar variability and volcanic activity responses (Schurer et al., 2013; Mann et al., 2009), with additional contributions from ocean variability (Jungclaus et al., 2010) and ice cover feedbacks (Miller et al., 2012). After the LIA, anthropogenic activities have a major impact on most of the observed temperature changes (Masson-Delmotte et al., 2013).

Temperature changes during the LM were noticeable at global scales, consistently with changes in the global energy balance (Crowley, 2000), and at continental and large regional scales (Tardif et al., 2019; Wilson et al., 2016; Anchukaitis et al., 2017).

Because of its direct dependence on the energy balance, the response to external forcing is evident in temperature. However, coordinated changes during the LM are also found in atmospheric dynamics and hydroclimate. In extratropical areas, changes in large-scale modes of variability like the North Atlantic Oscillation (NAO) have been found both in reconstructed and simulated

data (Cook et al., 2019; Trouet et al., 2009; Ortega et al., 2013). Even if for tropical areas larger uncertainties exist, long-term coordinated changes also have been found, mostly related to the variability of the El Niño - South Oscillation phenomenon (ENSO; Mann et al., 2009; Emile-Geay et al., 2013a, b).

An alteration of the circulation modes can also lead to changes in the hydroclimate of many regions. This has been found both in analyses based on reconstructed data, and to a lesser extent in analyses based on simulations (Ljungqvist et al., 2016).

Reconstructions of the hydroclimate of east Africa (Anchukaitis and Tierney, 2013), Mediterranean area (Luterbacher et al., 2012), western North America (Cook et al., 2010; Steinman et al., 2013) and tropical South America (Vuille et al., 2012) show coordinated changes in the hydroclimate of the LM. The spatial pattern associated with these changes is mainly latitudinal, with latitudes of increased and decreased precipitation during the MCA and the LIA. In particular, northern Europe and northern North America showed wetter conditions during the MCA and drier during the LIA, and conversely for southern Europe

and southern North America (Cook et al., 2010; Luterbacher et al., 2012). Tropical areas, like east Africa and tropical South America, also showed increased precipitation in the north during the MCA and in the south during the LIA (Anchukaitis and Tierney, 2013; Vuille et al., 2012).

Such coordinated changes in atmospheric dynamics and hydroclimate, with out-of-phase regional behavior during the MCA and LIA, are suggestive of large-scale responses to external forcing. One possibility is that changes in global temperatures, as

a consequence of changes in the forcing, could have altered modes of variability, and this could have had in turn an impact on hydroclimate (Zorita et al., 2005). The latitudinal distribution of these changes both in tropical and extratropical areas is suggestive of a mechanism based on displacements of the Intertropical Convergence Zone (ITCZ) and expansions and contractions of the Hadley Cells (Newton et al., 2006; Graham et al., 2010). These changes may have contributed to the alteration of modes of variability like the NAO and Northern Annular Mode (NAM) in the Northern Hemisphere (NH) and

Southern Annular Mode (SAM) in the Southern Hemisphere (SH).

To assess the influence of external forcing and internal variability on these changes, we perform analyses of simulations from the Coupled Model Intercomparison Project Phase 5 / Paleoclimate Modelling Intercomparison Project Phase 3 (CMIP5/PMIP3; Taylor et al., 2007; Stocker et al., 2013). The use of a large set of simulations generated with different models and with different boundary conditions increases the reliability of results and allows us to sample different climate sensitivities

and different plausible external forcing histories. Simulations from the Community Earth System Model - Last Millennium En-



**Table 1.** CMIP5/PMIP3 models analysed in this work, including the number of simulations considered in the analyses (NS), the resolution of the atmosphere and ocean components (latitude, longitude and levels), the forcing factors considered in the simulations (O = Orbital; S = Solar; V = Volcanic; G = Greenhouse gases; A = Aerosols; L = Land use and cover; according to Masson-Delmotte et al., 2013), and the references for the model experiments. All simulations span the period 850-2005 CE. This interval will be referred herein as LM, even if within PMIP3 LM only includes 850-1850 CE.

| Model | NS | Res. Atm. | Res. Ocean | Forcings | References |
|-------|----|-----------|------------|----------|------------|
| CSIRO | 1 | 3.2º x 5.6º - L18 | 1.6º x 2.8º - L21 | O, S, V, G | Phipps et al. (2012) |
| IPSL | 1 | 1.9º x 3.8º - L39 | 1.2º x 2º - L31 | O, S, V, G | Dufresne et al. (2013) |
| MRI | 1 | 1.9º x 3.8º - L40 | 0.5º x 1º - L50 | O, S, V, G | Yukimoto et al. (2011); Adachi et al. (2013) |
| MPI | 1 | 1.8º x 1.8º - L47 | 0.8º x 1.4º - L40 | O, S, V, G, A, L | Giorgetta et al. (2013) |
| CCSM | 1 | 1.25º x 0.9º - L26 | 1º x 1º - L60 | O, S, V, G, A, L | Landrum et al. (2013) |
| HadCM | 1 | 2.5º x 3.75º - L19 | 1.25º x 1.25º - L20 | O, S, V, G, A, L | Tett et al. (2007) |
| CESM | 13 | 2º x 2º - L26 | 1º x 1º - L60 | O, S, V, G, A, L | Otto-Bliesner et al. (2015) |
| GISS | 3 | 2º x 2.5º - L40 | 1º x 1.25º - L32 | O, S, V, G, A, L | Schmidt et al. (2006, 2014) |

semble (CESM-LME; Otto-Bliesner et al., 2015) also have been included. The use of the 13 simulations of the CESM-LME, using the same boundary conditions but with different initial conditions, allows for a systematic sampling of internal variability. With these simulations, the contribution of external forcing and internal variability to temperature, atmospheric circulation and hydroclimate has been analysed. The changes in the forcing factors during the LM incorporate both short-term changes, associated with individual volcanic events, and long-term changes, related to variations in solar activity and orbital orientation. Our analyses allow us to evaluate whether global responses in temperature, dynamics, and hydroclimate to external forcing are manifest in climate models, what is their expected spatial distribution, and whether the simulated responses are consistent with those obtained from reconstructed data.

## 2 Models and methods

Analyses are performed with simulations from the CMIP5/PMIP3, including three simulations from the Goddard Institute for Space Studies (GISS; Schmidt et al., 2006, 2014), individual LM simulations from the Community Climate System Model (CCSM; Landrum et al., 2013), the Hadley Centre Coupled Model (HadCM; Tett et al., 2007), the models of the Commonwealth Scientific and Industrial Research Organization (CSIRO; Phipps et al., 2012), the Institut Pierre Simon Laplace (IPSL; Dufresne et al., 2013), the Max Planck Institut für Meteorologie (MPI; Giorgetta et al., 2013), the Meteorological Research Institute (MRI; Yukimoto et al., 2011; Adachi et al., 2013), and with 13 additional simulations from the CESM-LME. Some technical details of these models are summarised in Table 1, including the horizontal and vertical resolutions, the number of simulations covering the last millennium and the forcings considered in these simulations.



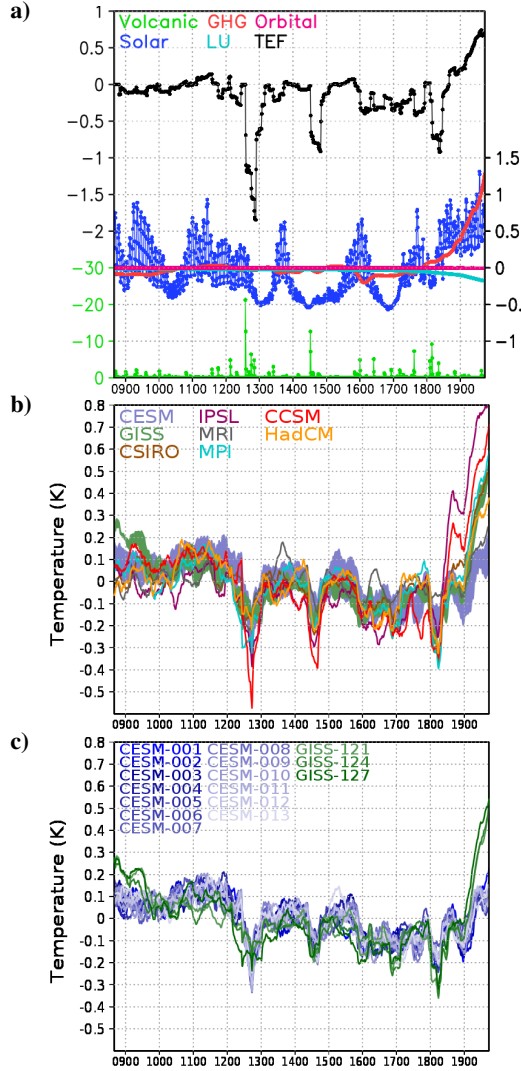

**Figure 1.** **(a)** External forcing factors in $Wm^{-2}$ considered in the CESM-LME simulations, including greenhouse gases (GHG), volcanic, solar, land use and cover (LU) and orbital forcing. The CESM-LME is selected as an example since it uses standard CMIP5/PMIP3 forcing specifications (Schmidt et al., 2011, 2012). The forcing related to greenhouse gases is obtained by composing the contributions of $CO_2$, $CH_4$ and $N_2O$. Volcanic forcing is represented with a different scale (in green). An evaluation of the Total External Forcing (TEF) following Fernández-Donado et al. (2013) is also included. **(b)** Global average of temperature in the ensemble of simulations. For CESM and GISS, the range of minimum-maximum values in the subsensemble at each time step is shown, instead of individual model simulations. **(c)** Individual simulations in CESM and GISS subensembles. All temperature and TEF series have been 31 years low-pass filtered with a centered moving average.

Even if ensembles with individual forcing factors and with reduced sets of forcings are available for some particular models, only simulations with the most complete set of external forcings that were available for each model have been considered



herein. As shown in Table 1, all the analysed simulations include solar variability, volcanic eruptions, greenhouse gases and orbital changes as external forcings, while most of them also include land use and cover changes (LULC) and anthropogenic aerosols. The reconstructions of forcings considered for each simulation can be found in Masson-Delmotte et al. (2013) for the CMIP5/PMIP3 simulations and in Otto-Bliesner et al. (2015) for the CESM-LME. The forcing factors are similar in all

the simulations, based on the guidelines for CMIP5/PMIP3 (Schmidt et al., 2011, 2012). The greenhouse gases are mainly obtained from MacFarling-Meure et al. (2006), the orbital changes from Berger (1978), the volcanic forcing from Gao et al. (2008) and Crowley and Unterman (2013), the solar variability from Vieira et al. (2011), Wang et al. (2005) and Steinhilber et al. (2009), the land use from Pongratz et al. (2008) and the aerosols from Lamarque et al. (2010).

Figure 1 plots the forcing factors considered in the CESM-LME simulations as an example of the LM forcings. The figure

also shows an estimation of the Total External Forcing (TEF) obtained by composing the contributions of several forcing factors according to Fernández-Donado et al. (2013). TEF shows the long-term evolution of the overall incoming energy with long intervals of higher (MCA and industrial era) and lower (LIA) forcing, as well as multidecadal to centennial changes produced by the combination of solar and volcanic activity.

The response to external forcing of temperature and a number of variables representative of atmospheric circulation and

hydroclimate conditions is analysed; namely sea level pressure (SLP), zonal wind (U), precipitation (P), soil moisture (SM), precipitation minus evaporation (P-E) and the Palmer Drought Severity Index (PDSI; Palmer, 1965), with the potential evapotranspiration calculated using the Thornthwaite's method (Thornthwaite, 1948). While this formulation of PDSI has been shown to be problematic for soil moisture assessments during projections of the 21st century, differences between different formulations of PDSI have been shown to be minimal over the LM interval even when the 20th century is included (Smerdon

et al., 2015). Characterising the hydrological state of the soil is challenging because different General Circulation Models (GCMs) incorporate different land models and soil moisture physics. This work is focused therefore on precipitation as well as P-E as descriptors of general surface hydrological interactions, and on soil moisture and PDSI as descriptors of soil moisture content. PDSI is incorporated to employ an homogeneous description of soil moisture content across models because it is calculated from atmospheric variables and assumes the same soil parameters across all of the models. It has been defined

following the self-calibrating PDSI index (scPDSI; Wells et al., 2004). All simulations have been interpolated to a common 6ºx6º grid resolution, the coarsest among the analysed simulations.

At interannual scales, some events of external forcing such as volcanic eruptions are able to explain large changes in the analysed variables (Fischer et al., 2007). To assess the impact of such events on the climate, we use a Superposed Epoch Analysis (SEA), by defining a composite with the five years before and ten years after the main volcanic eruptions within the

LM and computing for this composite the global average of the variables previously mentioned. Volcanic eruptions for this analysis have been selected following the procedure in Masson-Delmotte et al. (2013). For simulations of CESM-LME and CCSM, the reconstruction from Gao et al. (2008) has been considered, obtaining the minima of forcing in the years 1452, 1584, 1600, 1641, 1673, 1693, 1719, 1762, 1815, 1883, 1963 and 1990. For the other simulations, the reconstruction from Crowley and Unterman (2013) has been considered, generating a composite with the years 1442, 1456, 1600, 1641, 1674, 1696, 1816,

1835, 1884, 1903, 1983 and 1992.





**Table 2.** Correlations between the first PC of temperature of each model simulation and TEF (Fig. 1a) and correlations between the first PC of temperature and the PC linked to the forcing of each of the other selected variables (first PC of sea level pressure, zonal wind and scPDSI and second PC of precipitation, soil moisture and P-E) for different simulations. Significant correlations (p<0.05) are shown in bold.

| Simulation | T / TEF | SLP / T | U / T | P / T | SM / T | P-E / T | scPDSI / T |
|---|---|---|---|---|---|---|---|
| CSIRO | **0.72** | 0.26 | 0.33 | **0.78** | 0.21 | **0.65** | **0.87** |
| IPSL | **0.71** | 0.15 | 0.27 | **0.89** | 0.02 | **0.90** | - |
| MRI | **0.65** | 0.09 | 0.07 | **0.59** | 0.07 | 0.33 | - |
| MPI | **0.72** | **0.57** | **0.46** | **0.87** | 0.33 | **0.67** | **0.88** |
| CCSM | **0.79** | **0.68** | **0.41** | **0.91** | 0.03 | **0.77** | **0.90** |
| HadCM | **0.70** | **0.35** | **0.42** | **0.82** | 0.08 | **0.59** | - |
| CESM-001 | **0.61** | **0.36** | **0.54** | **0.59** | 0.14 | **0.45** | **0.68** |
| CESM-002 | **0.61** | **0.41** | 0.26 | **0.50** | 0.10 | 0.28 | **0.57** |
| CESM-003 | **0.66** | **0.46** | 0.24 | **0.66** | **0.39** | **0.39** | **0.51** |
| CESM-004 | **0.60** | 0.29 | 0.19 | **0.57** | 0.03 | 0.29 | **0.59** |
| CESM-005 | **0.61** | **0.43** | 0.17 | **0.56** | 0.07 | 0.33 | **0.61** |
| CESM-006 | **0.65** | **0.40** | 0.19 | **0.58** | 0.19 | **0.38** | **0.66** |
| CESM-007 | **0.66** | **0.40** | 0.27 | **0.62** | 0.14 | **0.38** | **0.63** |
| CESM-008 | **0.65** | 0.27 | 0.24 | **0.56** | 0.05 | **0.40** | **0.74** |
| CESM-009 | **0.61** | **0.36** | 0.13 | **0.55** | 0.15 | 0.33 | **0.70** |
| CESM-010 | **0.61** | **0.53** | 0.29 | **0.63** | 0.22 | 0.33 | **0.58** |
| CESM-011 | **0.72** | **0.49** | 0.25 | **0.62** | 0.09 | **0.36** | **0.67** |
| CESM-012 | **0.64** | 0.25 | 0.25 | **0.64** | 0.16 | **0.37** | **0.64** |
| CESM-013 | **0.63** | **0.50** | 0.28 | **0.64** | 0.19 | **0.41** | **0.66** |
| GISS-121 | **0.64** | **0.46** | 0.24 | **0.88** | 0.17 | **0.77** | **0.88** |
| GISS-124 | **0.67** | **0.69** | **0.56** | **0.91** | 0.21 | **0.76** | **0.80** |
| GISS-127 | **0.67** | **0.46** | 0.12 | **0.88** | 0.32 | **0.84** | **0.88** |
| Average | **0.73** | **0.59** | **0.51** | **0.81** | 0.20 | **0.62** | **0.94** |

We use Principal Component (PC) analyses to describe the spatiotemporal response of variables at multidecadal to centennial timescales. The analysis has been applied to data low-pass filtered with a moving average of 31 years, in order to emphasize the response to external forcing versus internal variability (Fernández-Donado et al., 2013). We concatenate all of the simulations to determine the average Empirical Orthogonal Functions (EOFs) across all of the models. The PCs are later obtained by
5   projecting each individual simulation on the EOF and, to assess the agreement among different simulations, the correlation coefficients between PCs from each simulation have been computed. This method allows us to obtain the modes of variability





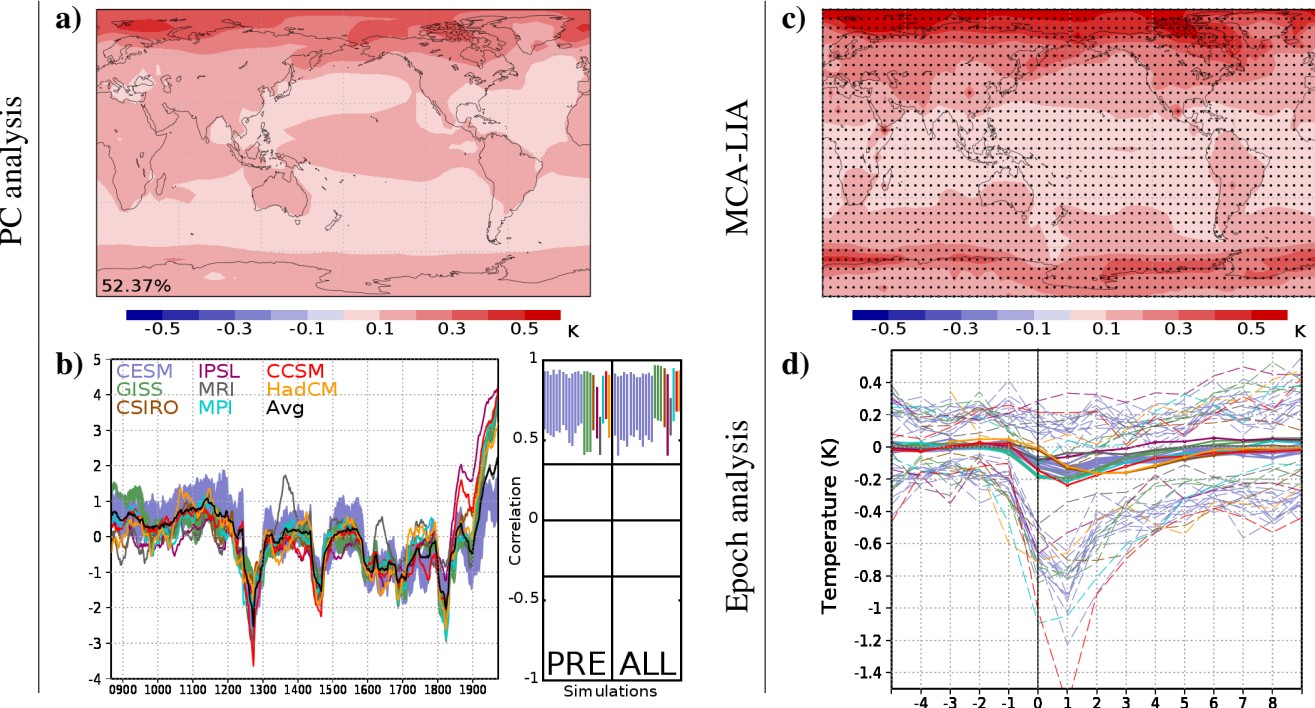

**Figure 2.** Analysis of temperature for the ensemble of simulations included in Table 1. **(a)** First EOF and **(b)** First PC time series for each simulation, as well as the average PC of all the simulations (black line). The percentage of explained variance is shown within the EOF map. The range of correlations between the PC of each simulation and those of other simulations is also included to the right of the plotted PCs, both for the whole period (ALL) and for the pre-industrial era (PRE). For these correlations, the significance level (p<0.05) is shown with a black line. **(c)** Map of temperature differences between MCA and LIA. Dots indicate locations where the differences are significant (p<0.05). **(d)** Composite average (solid line), and maxima and minima (dashed line) of global temperature anomalies in the five years before and ten years after the 12 main volcanic eruptions of the LM. Vertical line indicates the year of the volcanic event.

that explain a larger percentage of variance when considering the whole ensemble, because the covariability with external forcing is expected to show common features across models and simulations.

The results presented herein have been confirmed for consistency with individual analysis of each model simulation. The resulting EOFs bear only regional differences that do not contradict the results obtained with the combined analyses. The con-

5   catenation of some relatively large subensembles (GISS and CESM-LME) may bias the results to these models, but individual analyses confirm that is not the case. As an advantage, the concatenation of all simulations allows us to define EOFs that are valid for all models. Additionally, the incorporation of two subensembles allows for insights about the effects of internal variability, since the GISS and CESM ensembles incorporate identical boundary conditions and different initial conditions for each simulation.





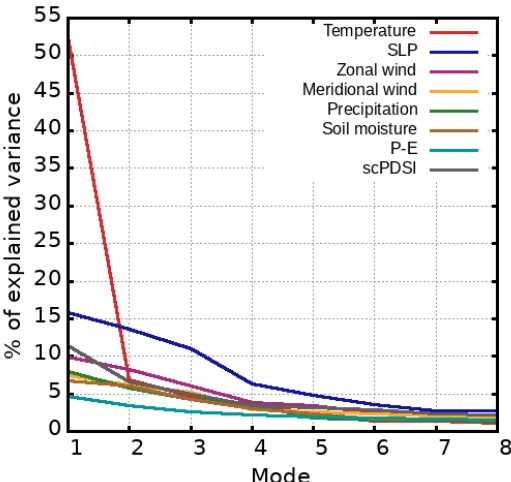

**Figure 3.** Percentage of variance explained by the first eight modes obtained with a PC analysis of temperature, SLP, zonal and meridional wind, precipitation, soil moisture, P-E and scPDSI.

Some long-term changes in the external forcing, like the one during the transition from MCA to LIA, are significant enough to be obtained not only by performing PC analyses but also by directly looking at the evolution of the variables during these two periods. To further analyse this transition, composites for the MCA and LIA have been defined from the ensemble average of each variable, using the years between 950 and 1250 CE and 1450 and 1850 CE, respectively. Composite maps of the differences between these two periods are derived for temperature, SLP, zonal wind, precipitation, soil moisture, P-E and scPDSI.

## 3 Results

### 3.1 Temperature

Analyses based on simulations and reconstructions show that temperature during the LM experienced an evolution consistent with the temporal character of the major forcings (Zorita et al., 2005; Fernández-Donado et al., 2013). The peaks in volcanic forcing after the main eruptions are related to periods with lower global temperatures, while the multidecadal variability and long-term trends associated with solar and anthropogenic forcings correspond with the long-term changes in temperatures that define periods of the MCA, LIA, and industrial era. This is found both at hemispheric and continental scales (Masson-Delmotte et al., 2013; Luterbacher et al., 2016; Büntgen et al., 2011; PAGES 2k Consortium, 2013), even if the temporal phasing of these periods are regionally dependent and depend on the specific reconstruction or model (Neukom et al., 2014, 2019). The temporal intervals of these periods are adopted from Masson-Delmotte et al. (2013), with the MCA ranging from 950 to 1250 CE, the LIA from 1450 to 1850 CE, and the industrial period after 1850 CE, although these time frames may not be regionally optimal.





Figure 1 shows global temperature averages for all the ensemble members listed in Table 1. As reported by Fernández-Donado et al. (2013) in the case of the pre-PMIP3 LM experiments, there is a quasi-linear response of temperatures to changes in external forcing, with the major warmings occurring in periods with high solar activity and GHG rise, and coolings occurring in response to lower solar forcing and increased volcanic activity. For the 20th century, all the analysed simulations consistently

show a warming, but trends strongly differ among simulations due to the different climate sensitivities of each model and the considered forcings. Simulations of IPSL, CCSM and GISS show a very strong trend, while simulations of CESM show a significant but smaller 20th-century temperature increase. The results for the GISS and CESM subensembles are shown in Fig. 1b by plotting the minimum-maximum spread of all subensemble members, while Fig. 1c shows, for the sake of clarity, the behavior of each subensemble member. These subensembles demonstrate that internal variability generates differences across

simulations that are smaller than the differences due to structural differences in model formulation across different models. In a related and most relevant note, changes in the ensemble associated with external forcing are in general more relevant than those of internal variability.

A similar behaviour is found when performing a PC analysis of the simulated temperatures, as shown in Fig. 2, with the first temperature EOF and PC estimates obtained from the simulations listed in Table 1. Agreement between the different

simulations is very good, all of them showing the same long-term trends described for Fig. 1. Figure 2 also shows the correlation among the estimated PCs of different simulations. Note that most of the analysed simulations show correlations larger than 0.5 and for simulations of the same model the correlations reach values around 0.9, both when analysing the whole period and when considering only the pre-industrial era. This indicates that even if the EOF has been obtained with a combined analysis, it is also representative of the individual simulations. Additionally, the use of large sets of simulations for some of the models, and

in particular the use of the 13 CESM-LME simulations, does not significantly bias the results, because the correlation ranges for models with individual simulations are as large as for the others. The variance of temperature that the first PC accounts for in the ensemble is larger than 50% across all of the models, significantly larger than the variance explained by the remaining modes (Fig. 3). To evaluate whether this first mode is related to the forcing, the correlations of the PC of each simulation with the TEF (Fig. 1a) have been included in Table 2. All the correlations are significant and range between 0.61 and 0.79. The first

PC of temperature is therefore mainly attributed to external forcing, and for this reason dynamics and hydroclimate variables are compared in the following sections against this mode to explore their forced responses.

Regarding the spatial pattern of the EOF, values are larger over the ice covered and continental regions and smaller over oceans. For high latitudes, this behaviour is known as the polar amplification response, and is consistent with that in climate change scenarios (Bindoff et al., 2013). For all regions the first EOF shows positive loadings, meaning that situations of higher

forcing correspond to larger temperatures for the whole planet and vice versa. This pattern has been discussed in Zorita et al. (2005) and Fernández-Donado et al. (2013), and can be also observed in the differences between the MCA and LIA (Fig. 2c). The transition from MCA to LIA in most regions leads to a decrease of simulated temperatures, consistent with the negative external forcing anomalies applied in the simulations, with a pattern of change very similar to that of the leading EOF (Fig. 2a).




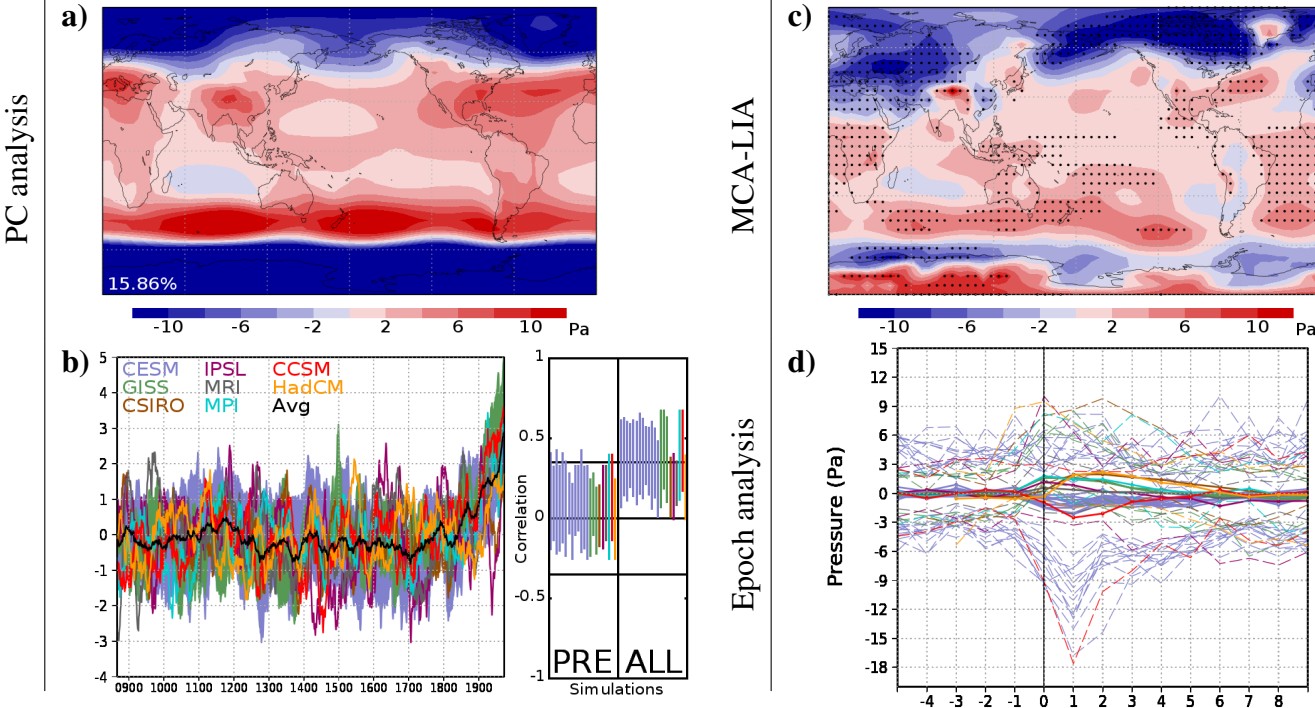

**Figure 4.** Analysis of SLP for the ensemble of simulations included in Table 1. **(a)** First EOF and **(b)** First PC time series for each simulation, as well as the average PC of all the simulations (black line). The percentage of explained variance is shown within the EOF map. The range of correlations between the PC of each simulation and those of other simulations is also included to the right of the plotted PCs, both for the whole period (ALL) and for the pre-industrial era (PRE). For these correlations, the significance level ($p<0.05$) is shown with a black line. **(c)** Map of SLP differences between MCA and LIA. Dots indicate locations where the differences are significant ($p<0.05$). **(d)** Composite average (solid line), and maxima and minima (dashed line) of global SLP anomalies in the five years before and ten years after the 12 main volcanic eruptions of the LM. Vertical line indicates the year of the volcanic event.

At shorter timescales, the influence of external forcing is also evident. Figure 2d shows the average, maxima and minima of temperature for the composite with the five years before and ten years after the 12 main volcanic eruptions of the LM. Note that global temperature significantly decreases after these events, and their influence lasts several years until it totally disappears. There exist important differences among models in the response to volcanic eruptions. Some simulations, such as those from
5    GISS, MPI, CESM and CCSM, show a large cooling after volcanic events that is mostly recovered after four years. However, simulations from HadCM and CSIRO show a moderate cooling that lasts for a longer period.

## 3.2   Atmospheric dynamics

The atmospheric circulation during the LM has been mainly analysed by focusing on modes of internal variability, such as Pacific Decadal Oscillation (PDO) and ENSO (e.g. Baek et al., 2017; Coats et al., 2016; Cook et al., 2019; Mann et al., 2009;





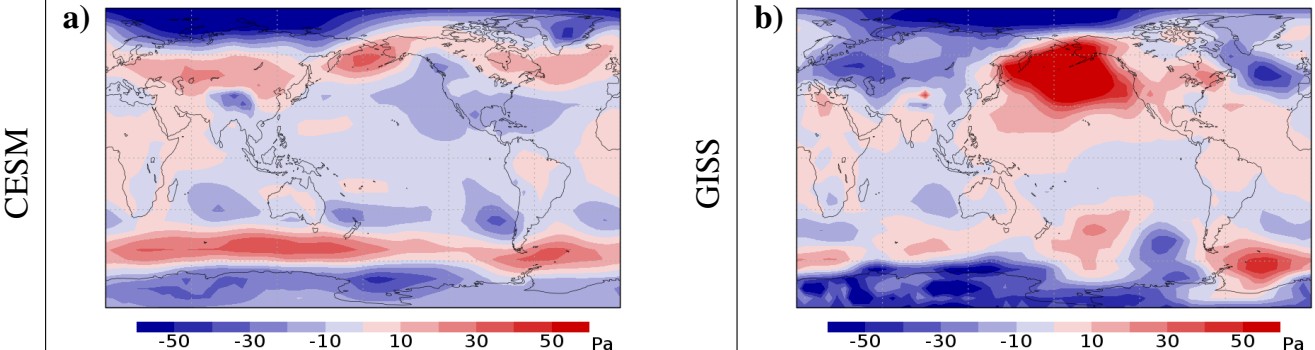

**Figure 5.** Maps of SLP anomalies for the 10 years after the 12 main volcanic eruptions of the LM. **(a)** Average of simulations from the CESM subensemble. **(b)** Average of simulations from the GISS subensemble.

Ortega et al., 2015; Emile-Geay et al., 2013a, b). Changes in external forcing are also likely to have had a significant impact on global atmospheric dynamics during the LM, for instance through influences on the position and size of the Hadley Cells, and therefore in the location and intensity of the large-scale modes of circulation in both hemispheres.

To evaluate some of these possible influences, the same analyses performed for temperatures have been applied to SLP.
Figure 4 shows results for the different timescales. The long-term behaviour of the first PC of pressure is similar to that of the first PC of temperature. For the case of pressure, higher values are also observed during the MCA, lower values during the LIA and a significant increase during the last century. This similarity can be quantified through the correlation coefficient values, that are significant for 73% of the simulations in the ensemble and above 0.5 for 23% of them. The average PC (black line in Fig. 4b) correlates with a value of 0.59 (Table 2) with the corresponding PC of temperature (black line in Fig. 2b).
This suggests some response to external forcing in the SLP field. The first mode accounts for only 15% of the variance in the pressure, showing an eigenvalue spectra flatter than the one of temperatures (Fig. 3). This indicates that even if there is a response to external forcing, internal variability is more relevant than in the case of temperature. Additionally, the correlations among PCs of different simulations are in general not significant over the pre-industrial period (Fig. 4b). When the whole period is considered, more significant correlations are obtained.
The EOF indicates that the first mode is mainly extratropical (Fig. 4a). In the SH, there is an increase of pressure (positive loadings) around 40º S and a decrease (negative loadings) around 80º S during the MCA (higher PC values), and conversely during the LIA. This pattern is related to the zonal SLP stratification produced by the SAM (Jones et al., 2009; Fogt et al., 2009), that responds to the long-term changes in external forcing consistently with the response in future climate change scenarios (Zorita et al., 2005; Stocker et al., 2013). Regarding the NH, the pattern of pressure associated with the first mode is not as zonal as in the SH, mostly because of the interaction with the continents. Overall, higher positive (negative) loadings
distribute over subtropical (polar) regions contributing to increase (decrease) the zonal flow during the MCA and industrial





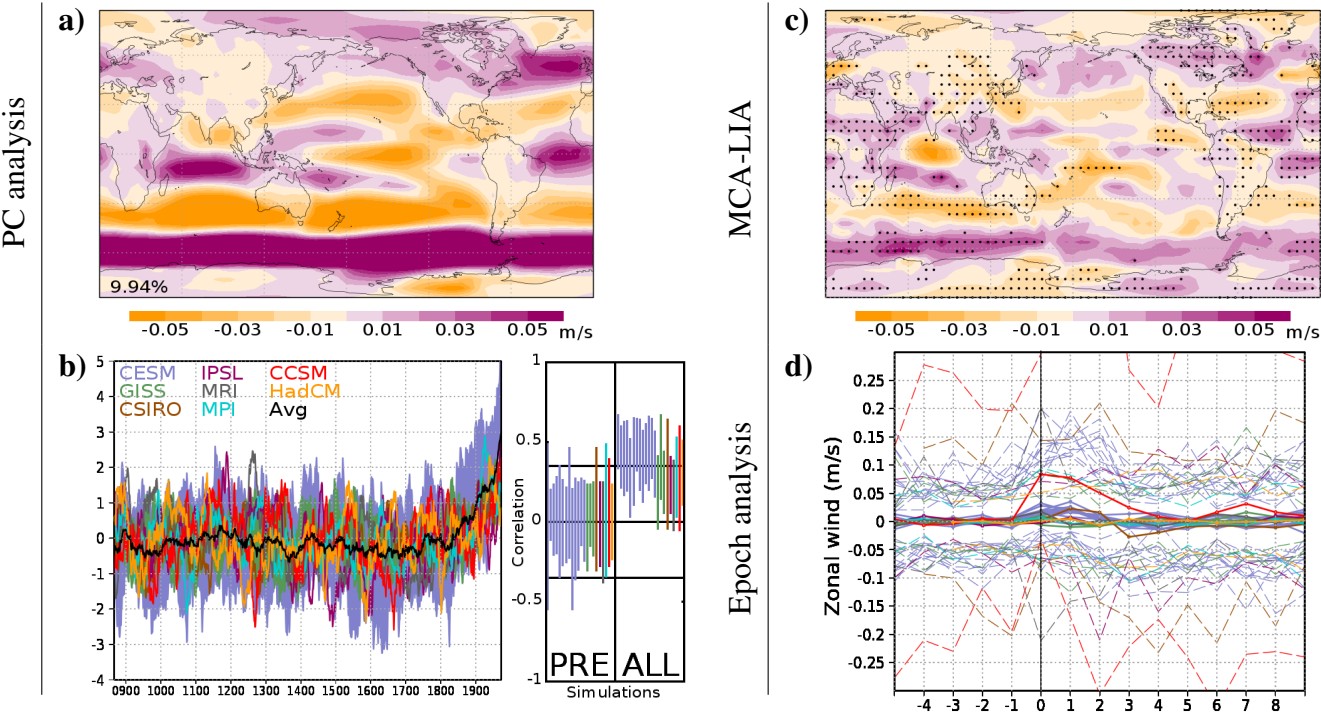

**Figure 6.** Analysis of zonal wind for the ensemble of simulations included in Table 1. **(a)** First EOF and **(b)** First PC time series for each simulation, as well as the average PC of all the simulations (black line). The percentage of explained variance is shown within the EOF map. The range of correlations between the PC of each simulation and those of other simulations is also included to the right of the plotted PCs, both for the whole period (ALL) and for the pre-industrial era (PRE). For these correlations, the significance level (p<0.05) is shown with a black line. **(c)** Map of zonal wind differences between MCA and LIA. Dots indicate locations where the differences are significant (p<0.05). **(d)** Composite average (solid line), and maxima and minima (dashed line) of global zonal wind anomalies in the five years before and ten years after the 12 main volcanic eruptions of the LM. Vertical line indicates the year of the volcanic event.

period, due to the slightly higher values of the PC. Therefore, a tendency toward more positive phases of the NAO, NAM and SAM is observed during the MCA and industrial periods.

A similar spatial pattern to that of the leading EOF is derived from a difference between the MCA and LIA periods (Fig. 4c). A latitudinal distribution of positive and negative differences is observed, related to the intensification (MCA) and weakening

5 (LIA) of the SAM and NAM/NAO. Even if the exact boundaries between areas with positive and negative MCA-LIA differences do not fully match the zonal features of the first EOF loadings, the ensemble average shows the largest differences over the subpolar and subtropical regions, which contribute to enhance the zonal flow in the MCA and weaken it during the LIA.

Due to the large range of internal variability, the short-term response to volcanic forcing is not so evident in the PC of SLP as it is in the case of temperature. However, some low frequency signal is observed in the mean PC series (Fig. 4b), for which high

10 frequencies are cancelled out. If the largest volcanic events are considered, a clearer response emerges. Figure 4d shows the





global average of SLP for the composites with the five years before and ten years after the main volcanic eruptions of the LM. Note that the global average of SLP shows visible changes triggered by volcanic activity. The sign and magnitude of the global net changes strongly depend on the model, but there is a good agreement among simulations of the same model. For instance, simulations of GISS show an increase of pressure after volcanic events, while simulations of CESM-LME consistently show

a decrease. This difference in the global average of pressure is not related to an opposite response in different models, but to the distribution of areas with positive and negative loadings in the mode of variability associated with the forcing. As shown in Fig. 5, simulations of CESM show a larger amount of areas with negative anomalies during periods with volcanic events, while simulations of GISS tend to show more areas with positive anomalies. In spite of the differences in the global balance of regional positive and negative anomalies among models, all of them produce a global weakening in zonal circulation during

volcanic eruptions.

In general, this global analysis shows that regional modes of variability might be indirectly influenced by external forcing, through changes in the distribution of pressure at global scales that interact with orography and physical properties at regional and local scales. Some studies have shown that elevated forcings during the 20th century may be linked to a displacement of the ITCZ and an expansion of Hadley Cells (Lu et al., 2007; Seager et al., 2007b). These changes in the general circulation

have also occurred during the MCA and LIA, and could explain the distribution of pressures in response to external forcing observed through the PC analysis and the analysis of composites. Increases in radiative forcing can produce a global warming and a significant intensification of latitudinal gradients of temperature; this intensification can generate an expansion and intensification of the Hadley Cells and a displacement of the ITCZ (Sachs et al., 2009); this expansion can further contribute to higher subtropical and lower subpolar pressures, thereby contributing to positive SAM and NAO/NAM phases.

The position of the ITCZ and the Hadley Cells can be described in terms of wind velocity, in which the former is defined by the location of the Trade winds and the extension of the latter marked by the position of the Westerlies (Frierson et al., 2007). Thus, the previous analysis is also performed for the zonal wind (Fig. 6). In this case, the first mode accounts for 10% of the total variance, a smaller percentage than for the case of temperature and SLP. The average PC of zonal wind shows a correlation of 0.51 with the average PC of temperature and 0.96 with that of SLP, indicating a response to the external forcing

consistent with the one observed in temperature and SLP. However, the correlation with the PC of temperature for the individual simulations is in general small, indicating a larger impact of internal variability. As shown in Table 2, only five simulations show significant correlations, and only two of them are larger than 0.5. The agreement among different simulations can be also assessed through the correlations among their respective PCs, which are in general not significant when the pre-industrial period is considered, and are only significant for certain simulations when they are based on the whole period (Fig. 6b). The

correlation between the corresponding PC time series of zonal wind and that of SLP for each simulation is always high and above 0.7 ($p<0.05$).

Thus, the distribution of zonal wind in the first EOF is consistent with the distribution of pressures previously shown. In situations of high forcing, such as during the MCA and the 20th century, the wind at around 30º N and 30º S decreases while the wind at around 60º N and 60º S increases (Fig. 6a). This suggests a poleward displacement of the Westerlies, consistent with

the expansion of the Hadley Cells observed in the SLP analysis. The same behaviour can be observed in the composites for





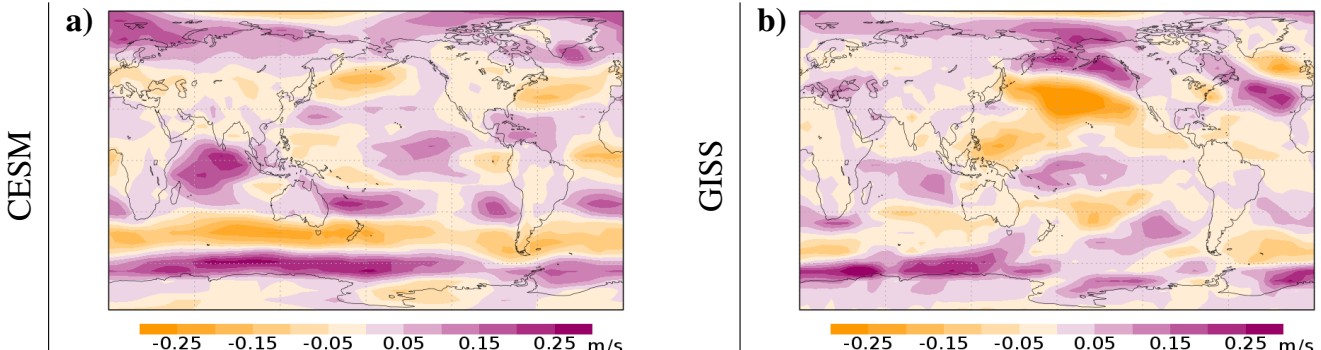

**Figure 7.** Maps of zonal wind anomalies for the 10 years after the 12 main volcanic eruptions of the LM. **(a)** Average of simulations from the CESM subensemble. **(b)** Average of simulations from the GISS subensemble.

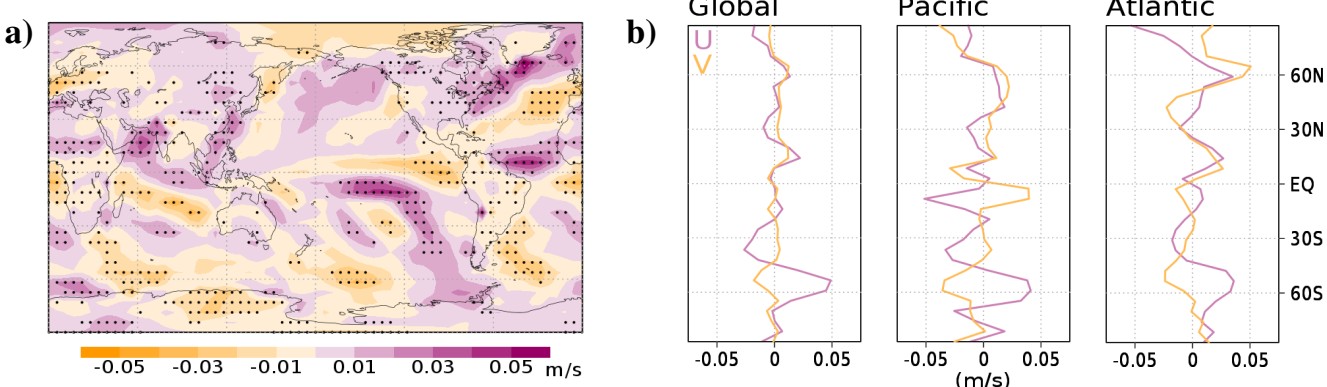

**Figure 8. (a)** Map of meridional wind differences between MCA and LIA. Dots indicate locations where the differences are significant (p<0.05). **(b)** Zonal mean of zonal wind (purple) and meridional wind (yellow) for the composite MCA-LIA, for global, central Pacific (180º W-120º W) and Atlantic (50º W-0º) basins.

the MCA and LIA, where the transition from the MCA to LIA is characterised by a poleward displacement of the Westerlies. However, the spatial pattern of the difference between the MCA and LIA differs for some regions relative to the one obtained from the EOF. For example, the differences between MCA and LIA indicate a reduction of zonal wind in the Mediterranean basin and an increase over Japan in the transition from MCA to LIA (Fig. 6c), while the loading in the EOF for these particular

5  areas (Fig. 6a) is negative and positive, respectively.

In the short term, the behavior of zonal wind resembles that of SLP. In most simulations, the short-term noise associated with internal variability dominates over the peaks associated with volcanic eruptions, as can be observed in Fig. 6b. When composites of the main volcanic eruptions are defined (Fig. 6d), a clear influence of volcanic activity emerges. As for the



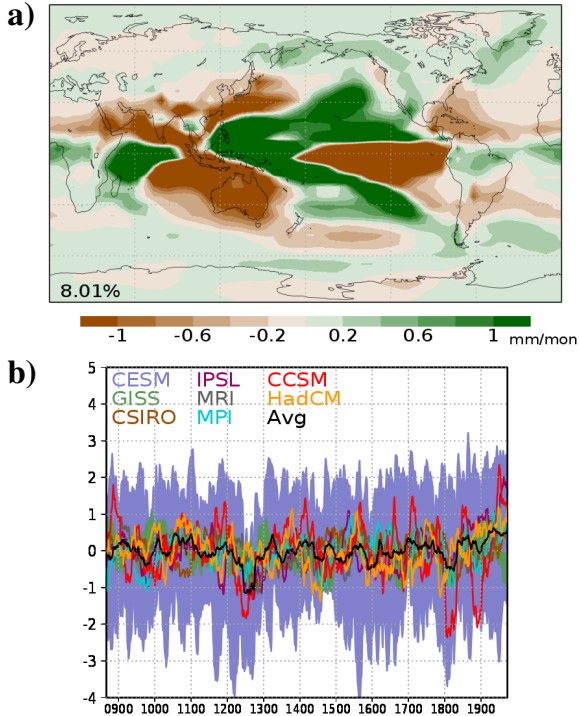

**Figure 9. (a)** First EOF of precipitation for the ensemble of simulations included in Table 1. The percentage of explained variance is shown within the EOF map. **(b)** PC time series for each simulation. The average of all the simulations is also included (black line).

case of pressures, the sign of the impact depends on the model, mostly because of the different spatial distribution of areas with positive and negative anomalies. This can be also observed when comparing maps of zonal wind during volcanic events obtained with the CESM and GISS subensembles (Fig. 7). The spatial patterns obtained with both subensembles are similar, although the simulations of CESM (GISS) show more areas with positive (negative) zonal wind, which translate into a larger

5    (smaller) increase of the global average. For all model simulations, the pattern tends to weaken the global zonal circulation.

These zonal changes are also evident in the behavior of the meridional wind component. Figure 8 shows the differences between MCA and LIA in terms of meridional wind, as well as the zonal mean of these differences for zonal and meridional components, both for global and regional Pacific (180º W-120º W) and Atlantic (50º W-0º) basins (Fig. 8b). During the simulated MCA, changes in zonal and meridional winds took place with anti phase relationships that strengthened the NAM and

10   SAM zonal circulation at mid and high latitudes, while within the intertropical regions the Trade winds and convergence were intensified (Fig. 6c and Fig. 8).





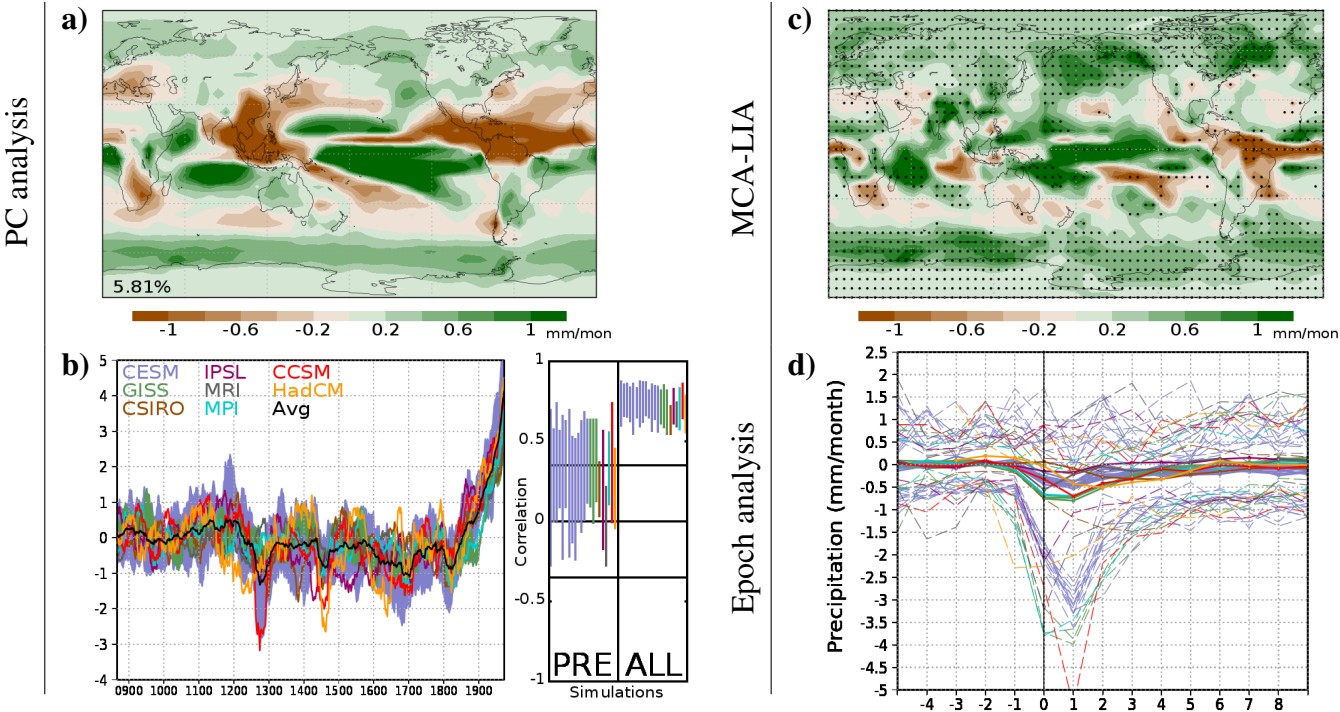

**Figure 10.** Analysis of precipitation for the ensemble of simulations included in Table 1. **(a)** Second EOF and **(b)** Second PC time series for each simulation, as well as the average PC of all the simulations (black line). The percentage of explained variance is shown within the EOF map. The range of correlations between the PC of each simulation and those of other simulations is also included to the right of the plotted PCs, both for the whole period (ALL) and for the pre-industrial era (PRE). For these correlations, the significance level (p<0.05) is shown with a black line. **(c)** Map of precipitation differences between MCA and LIA. Dots indicate locations where the differences are significant (p<0.05). **(d)** Composite average (solid line), and maxima and minima (dashed line) of global precipitation anomalies in the five years before and ten years after the 12 main volcanic eruptions of the LM. Vertical line indicates the year of the volcanic event.

### 3.3 Hydroclimate

Previous studies related to modes of variability and teleconnections have shown that variations in the distribution of atmospheric pressure may impact the amount of precipitation over large regions (Graham et al., 2007; Seager et al., 2007a; Feng and Hu, 2008). Thus, as the external forcing has been shown to play an important role in changing the atmospheric dynamics during
5  the LM (Sect. 3.2), it may also have a consistent influence on hydroclimate. The latter can be assessed from the same analyses performed for temperature, pressure and wind including variables of hydroclimate such as precipitation, soil moisture, scPDSI and P-E.

For the case of precipitation, Fig. 9 and Fig. 10 show the EOF and PC series for the leading two modes. Both modes show cross-basin influences in the tropical regions that connect the Indian, Pacific and Atlantic Oceans. Precipitation loadings are

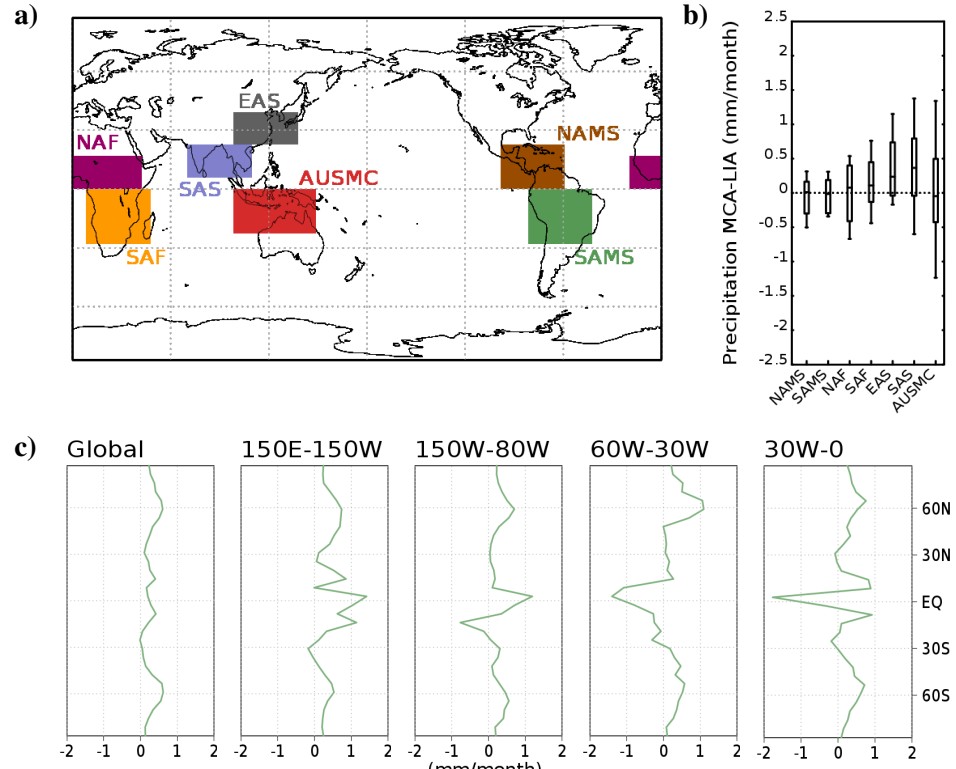

**Figure 11. (a)** Areas considered in the analysis of monsoon domains: North American Monsoon System (NAMS), South American Monsoon System (SAMS), North Africa (NAF), South Africa (SAF), East Asian Summer Monsoon (EAS), Southern Asian Summer Monsoon (SAS) and Australian-Maritime Continent (AUSMC). **(b)** Precipitation differences for MCA-LIA over the monsoon domains. Box and whisker plots show the 10th, 25th, 50th, 75th and 90th percentiles of simulations included in Table 1. **(c)** Zonal mean of precipitation for the composite MCA-LIA, for the whole globe and for bands of 150º E-150º W, 150º W-80º W, 60º W-30º W and 30º W-0º.

suggestive of ENSO, PDO and monsoon influences (Christensen et al., 2013), thus showing large-scale coordinated responses that can connect in-phase or out-of-phase intra and inter-continent responses. Although the first PC shows large negative values during the volcanically active 13th Century and some slight increase during the 20th century, it is the second PC that resembles the external forcing response. Furthermore, its EOF emphasizes the distribution of monsoonal precipitation over the global

5    monsoon domain within the intertropical region. In the extratropics the EOF pattern (Fig. 10a) shows distributions of loadings that are consistent with changes in NH NAM and SH SAM circulation. The PC series (Fig. 10b) clearly shows the long-term trends associated with the external forcing, with higher values during the MCA and 20th century and lower values during the LIA, as well as the short-term response to volcanic events, in agreement with the analysis of pressures (Fig. 4) and winds (Fig. 6).



**Figure 12.** Regions showing high positive (left) or negative (right) correlation between precipitation and the mode of precipitation associated to the forcing for the ensemble of simulations included in Table 1. Contours of correlation equal to -0.6 and 0.6 are shown for each simulation. Dots indicate locations where the time series of Fig.13 have been extracted.

There is a good agreement between different models and different simulations, with correlations larger than 0.5 for simulations of different models and reaching 0.9 for different simulations of the same model (Fig. 10b; ALL). The correlation between the second mode of precipitation and the first mode of temperature (Table 2) is significant for all the analysed simulations, ranging from 0.5 to 1, being higher in the simulations of GISS, IPSL and MPI and lower in the simulations of CESM and

MRI. These values are larger than those obtained for pressure and winds. However, this mode accounts only for 5.8% of the precipitation variance, a lower value than the one obtained for variables of dynamics. As seen in Fig. 3, the modes associated with internal variability acquire a larger relevance in the case of precipitation. This suggests that potential detection of this signal in reconstructions would be difficult and if possible, more likely at very local scales.

The MCA-LIA differences (Fig. 10c) show a similar distribution to the EOF loadings (Fig. 10a), with some regions receiving

more and others less precipitation when the forcing is higher. Changes in dynamics described in Sect. 3.2 are consistent with the changes in precipitation. As for the case of dynamics, some zonal symmetry is observed in extratropical areas, suggesting that changes in the northern and southern annular modes affect the distribution of precipitation in these regions. The highest variability occurs in intertropical areas, over the regions with the largest amount of annual rainfall, and thus overlapping well with changes in the global monsoon domain and ITCZ convergence. MCA-LIA differences show positive and negative rainfall

anomalies over the North and South American Monsoon Systems (NAMS, SAMS; Cerezo-Mota et al., 2011; Christensen et al., 2013). This agrees with uncertainty in climate change projections over the North American Monsoon region, with CMIP5 models producing changes in precipitation that distribute around zero. The same occurs over the Australian and Marine Continent Monsoon Systems (AMSMC; Jourdain et al., 2013). MCA-LIA differences show positive values over the East Asia and Southern Asian Summer Monsoon areas (EAS, SAS; May, 2011; Boo et al., 2011), in agreement with scenario

simulations (Christensen et al., 2013). Even if changes are not significant over many of these regions due to the large variability





**Figure 13.** Time series of precipitation in mm/month for some particular case example locations. The range of correlations between the time series of each simulation and those of other simulations at those specific locations is also included, both for the whole period (right; ALL) and for the pre-industrial era (left; PRE). For these correlations, the significance level (p<0.05) is shown with a black line. All precipitation series have been 31 years low-pass filtered with a centered moving average.

of precipitation, they show a consistent pattern of response to forcing for the current generation of climate models in LM PMIP3 simulations. Consistency also extends to convergence zones. In the Atlantic and eastern Pacific north of the Equator (e.g. Xie et al., 2007), negative MCA-LIA differences suggest increases in mean precipitation during the LIA. Likewise, in the South Pacific Convergence Zone extending from the western Pacific southeastwards (e.g. Widlansky et al., 2011), negative





differences also emerge, indicating increased rainfall. Over South America, rainfall shifts southeastwards and eastwards to the South Atlantic depicting changes in the South Atlantic Convergence Zone (Cavalcanti and Shimizu, 2012).

To better analyse these changes, MCA-LIA precipitation differences are calculated for the monsoon domains in Fig. 11a. Changes in precipitation in the transition from the MCA to LIA (Fig. 11b) are small in NAMS and SAMS for all the analysed

simulations, being slightly positive (negative) for NAMS (SAMS). These changes are consistent with a displacement of the convergence zone over the Americas, but important differences exist among models. Model simulations show larger discrepancies over Africa, where a clear difference between south and north is not found. The largest impact of MCA-to-LIA transition in the monsoon systems appears over Asia and Australia, where EAS, SAS and AMSMC are significantly altered. Rainfall anomalies in EAS and SAS monsoon areas are larger during the MCA relative to the LIA. This pattern is consistently found

in most of the simulations. For the AMSMC, some simulations show very positive differences while others are very negative, indicating that models all show important variations over this area but the magnitude and spatial distribution of these changes are strongly model dependent. The global zonal mean of precipitation for the MCA-LIA (Fig. 11c) does not show important changes but only slightly larger values of precipitation anomalies during the MCA than during the LIA for most latitudes. However, if ranges of longitude over the Pacific and Atlantic basins are considered, relevant changes in the convergence areas

are observed. The zonal mean for the range of 150º E-150º W shows larger precipitation rates during the MCA relative to the LIA in equatorial areas of the central and western Pacific. In the eastern Pacific (150º W-80º W), precipitation is decreased (increased) north (south) of the Equator during the MCA-to-LIA transition, suggesting a southward displacement of the convergence zone. Ranges of 60º W-30º W and 30º W-0º show that convergence zone is also altered in the Atlantic basin, with less intense precipitation around the Equator during the MCA than during the LIA.

Figure 10d shows the global average of precipitation anomalies in the five years before and ten years after the 12 main volcanic eruptions of the LM. A consistent decrease of global precipitation can be clearly observed in all of the simulations after these volcanic events. Consistent with changes in scenario simulations (Christensen et al., 2013), increases in external forcing strengthen the hydrological cycle, enhancing zonal circulation in extratropical regions and increasing the global monsoon activity and equatorial convergence. This is found in the global average of precipitation after volcanic events and in the

alteration of monsoons and latitudinal distribution obtained in the EOF, indicating a relevant response to external forcing in precipitation.

Figure 12 explores the robustness of regional forcing influences on precipitation across models and simulations. It shows areas of high positive correlation (above 0.6, $p<0.05$; left panel) and regions of high negative correlation (below -0.6, $p<0.05$; right panel) between precipitation and the mode of precipitation associated with the forcing of the ensemble of model simu-

lations. All models correlate strongly with external forcing over the same large-scale regions: the high-latitude bands related to zonal NAM and SAM circulation changes (e.g. negative correlations in the south of Europe and positive in the north) and the intertropical regions related to monsoon domains and convergence. Some models show larger areas of sensitivity to forcing while other present less regions where precipitation relates to the long-term changes in temperature produced by external forcing. For example, simulations of GISS show negative correlations in the north of Australia and the south of Africa that

do not appear in simulations of CESM-LME. This shows that regardless of the agreement in the big picture, the areas of high





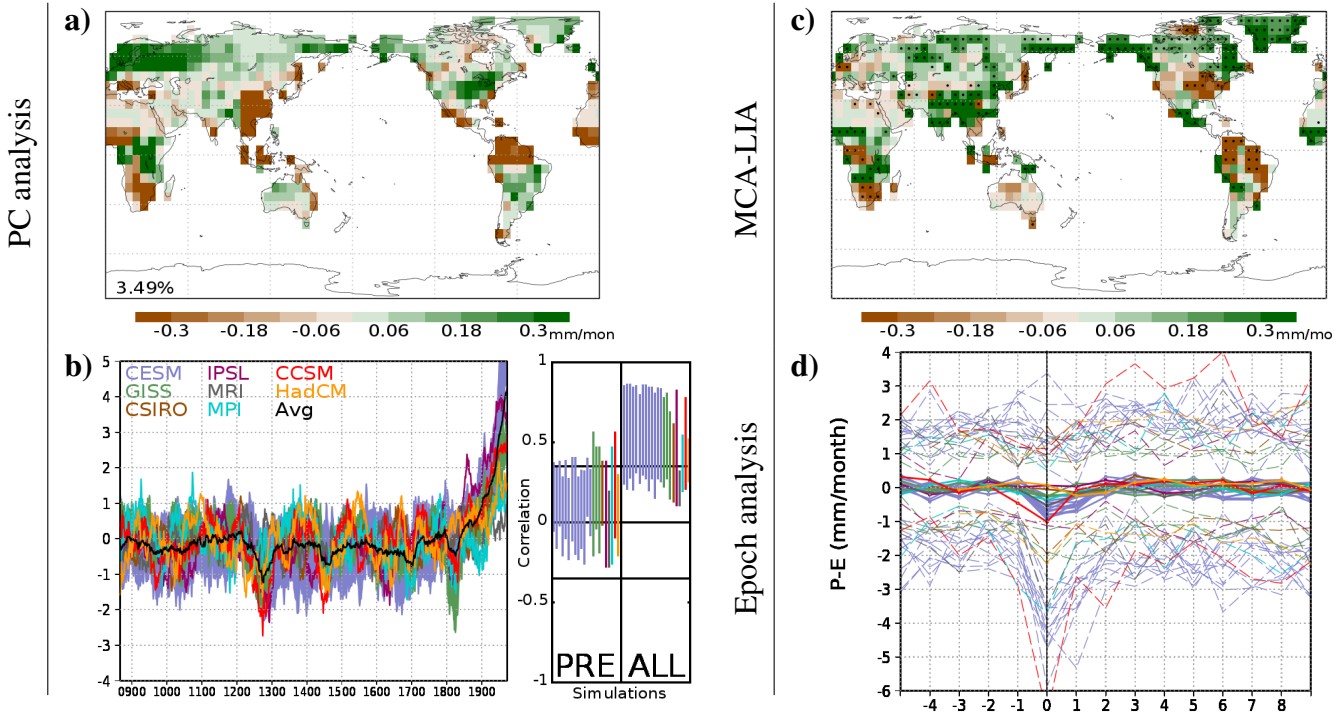

**Figure 14.** Analysis of P-E for the ensemble of simulations included in Table 1. **(a)** Second EOF and **(b)** Second PC time series for each simulation, as well as the average PC of all the simulations (black line). The percentage of explained variance is shown within the EOF map. The range of correlations between the PC of each simulation and those of other simulations is also included to the right of the plotted PCs, both for the whole period (ALL) and for the pre-industrial era (PRE). For these correlations, the significance level (p<0.05) is shown with a black line. **(c)** Map of P-E differences between MCA and LIA. Dots indicate locations where the differences are significant (p<0.05). **(d)** Composite average (solid line), and maxima and minima (dashed line) of global P-E anomalies in the five years before and ten years after the 12 main volcanic eruptions of the LM. Vertical line indicates the year of the volcanic event.

correlation are spatially very constrained to regional and even local scales and may not overlap in different models or even in simulations of the same model, this being a sign of the influence of internal variability. Correlations in high-latitude bands are all positive while negative correlations arise mostly over the areas of monsoon activity in Africa, Asia, and America (Fig. 12 right), as in the case of negative MCA-LIA anomalies discussed above. This view has implications for detection of a potential

5    response to global temperature and forcing changes in drought sensitive proxy data (e.g. Ljungqvist et al., 2016), as the spatial dimension of the response can be quite limited, particularly over land. Figure 13 shows examples of simulated precipitation at the gridpoint level in cases that tend to show some correlation with forcing like Spain, China, and Peru (see gridpoint locations highlighted with points in Fig. 12) and also at locations that tend to show limited connections to forcing within the ensemble: California, Quebec, and Brazil. At all these points the different ranges of internal variability tend to be larger than the forcing





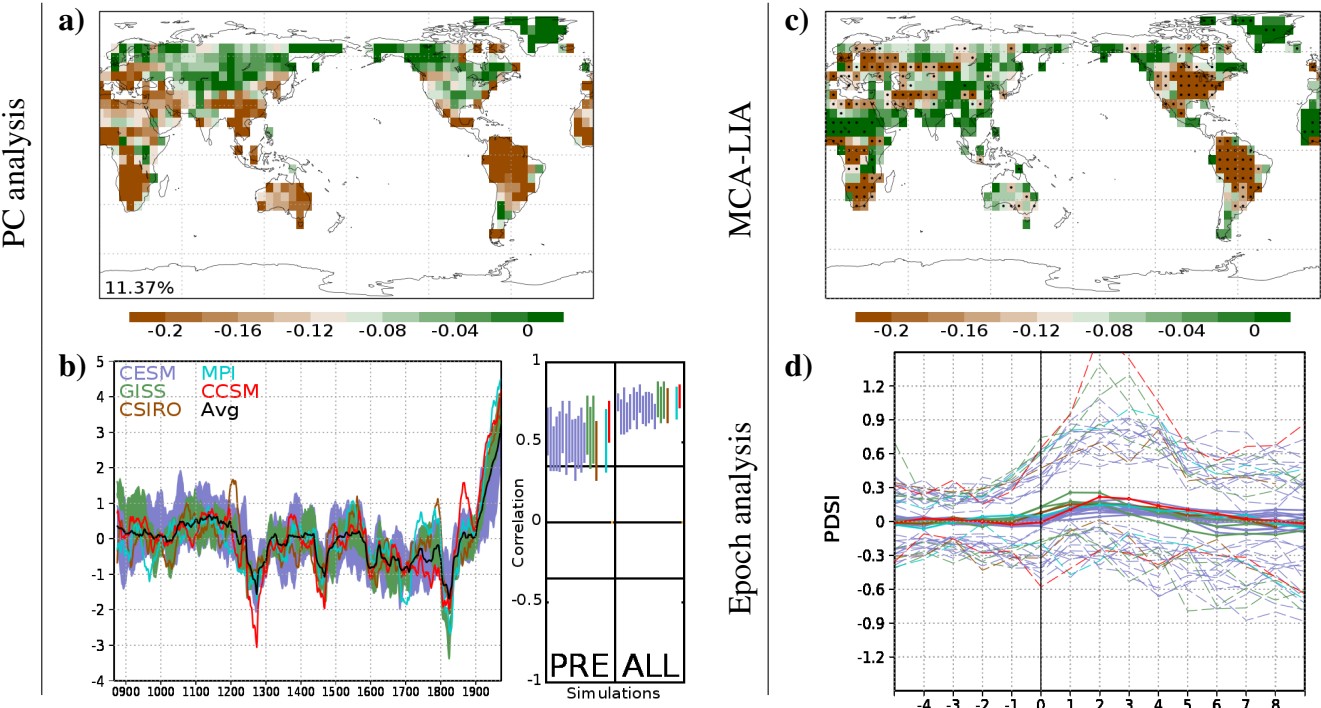

**Figure 15.** Analysis of scPDSI for the ensemble of simulations included in Table 1. **(a)** First EOF and **(b)** First PC time series for each simulation, as well as the average PC of all the simulations (black line). The percentage of explained variance is shown within the EOF map. The range of correlations between the PC of each simulation and those of other simulations is also included to the right of the plotted PCs, both for the whole period (ALL) and for the pre-industrial era (PRE). For these correlations, the significance level ($p<0.05$) is shown with a black line. **(c)** Map of scPDSI differences between MCA and LIA. Dots indicate locations where the differences are significant ($p<0.05$). **(d)** Composite average (solid line), and maxima and minima (dashed line) of global scPDSI anomalies in the five years before and ten years after the 12 main volcanic eruptions of the LM. Vertical line indicates the year of the volcanic event.

signal. In some areas the 20th-century trends show precipitation increases, like in the selected Brazil and Peru sites, while in others there is no response (e.g. the California site) or precipitation tends to decrease (e.g. the selected sites in Spain and China). Inter-simulation correlations tend to increase during the 20th century for sites showing trends and remain insignificant for most inter-ensemble pairs in pre-industrial times.

5     Similar results to those obtained for precipitation are obtained when analysing other variables representative of the water content of the soil. In particular, P-E, scPDSI and soil moisture have been analysed in this work. Even if these variables provide similar information, there exist important differences between them. Because soil moisture takes into account the water balance in previous time steps, together with precipitation, evaporation and temperature, its variations are typically smoother than those observed in atmospheric variables. Something similar happens with scPDSI, which similarly takes into account soil moisture

10     conditions from the previous months. Due to the higher number of factors considered in the computation of soil moisture and



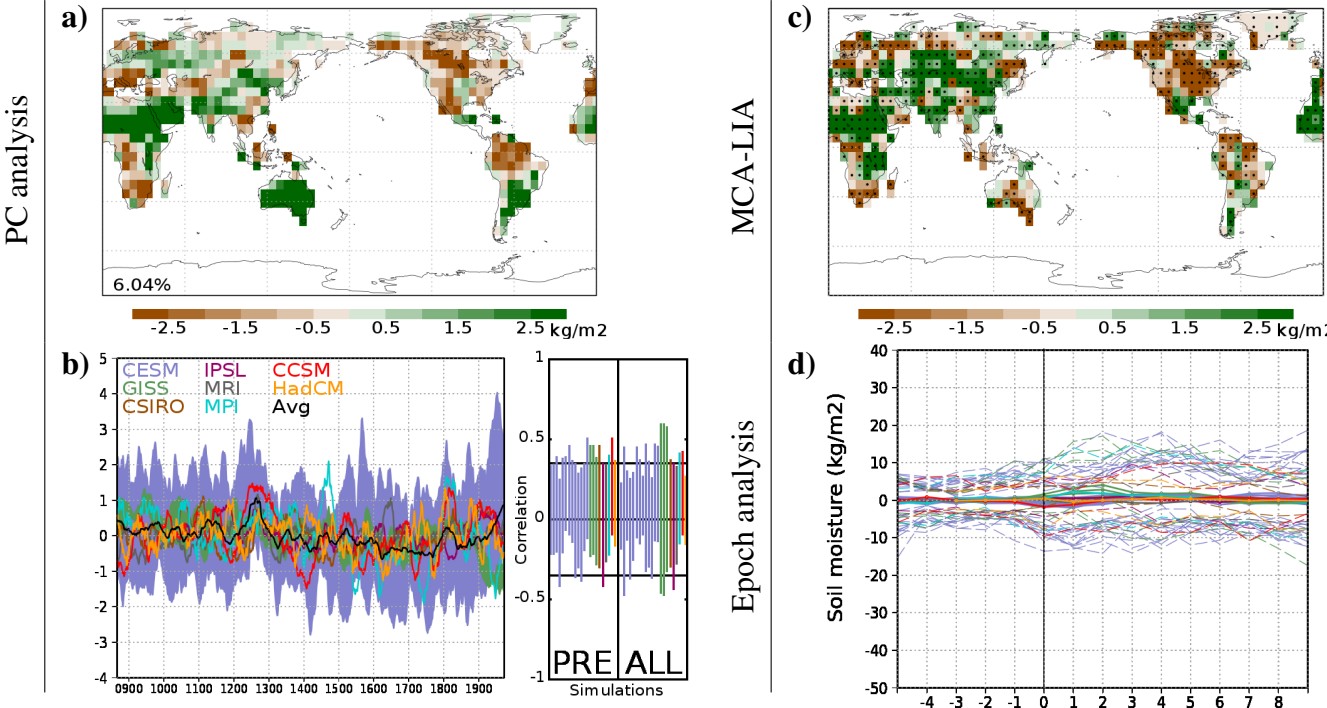

**Figure 16.** Analysis of soil moisture for the ensemble of simulations included in Table 1. **(a)** Second EOF and **(b)** Second PC time series for each simulation, as well as the average PC of all the simulations (black line). The percentage of explained variance is shown within the EOF map. The range of correlations between the PC of each simulation and those of other simulations is also included to the right of the plotted PCs, both for the whole period (ALL) and for the pre-industrial era (PRE). For these correlations, the significance level (p<0.05) is shown with a black line. **(c)** Map of soil moisture differences between MCA and LIA. Dots indicate locations where the differences are significant (p<0.05). **(d)** Composite average (solid line), and maxima and minima (dashed line) of global soil moisture anomalies in the five years before and ten years after the 12 main volcanic eruptions of the LM. Vertical line indicates the year of the volcanic event.

the fact that this variable is fully integrated within the models, it is a preferred variable for assessing drought variability in the models. However, different GCMs and their respective soil models may provide different treatments of soil moisture, making it difficult to combine data from different models in a single analysis. For this reason, the use of indices, such as scPDSI, may provide more consistent comparison across different models (e.g. Cook et al., 2014).

5    Figures 14 to 16 show the modes of P-E, scPDSI and soil moisture that bear relationship with global temperature and thus, external forcing responses. The associated mode is the second for P-E (Fig. 14) and soil moisture (Fig. 16) and the first for scPDSI (Fig. 15). P-E and scPDSI present PCs that correlate significantly with the temperature response mode (Fig. 2, Table 2); mean correlation values being 0.62 and 0.94 for P-E and scPDSI, respectively. Individual correlations are significant for most simulations and particularly high for those of GISS, IPSL and CCSM. The highest correlations are attained for scPDSI, being

10    significant and above 0.5 for all simulations, while for soil moisture correlations are smaller and in general not significant.





For P-E and scPDSI there is a clear time response pattern that shows an evolution very similar to that of precipitation (Fig. 10) and temperature (Fig. 2). In the case of P-E, the EOF spatial pattern is similar to that of precipitation (Fig. 10a), showing large negative values over parts of the central American and north of the South American monsoon regions, as well as over the East Asian continent and over parts of the African monsoon regions, where the EOF of precipitation also includes negative

loadings. Positive loadings are shown over extratropical continents, in northern Europe, southeastern North America and Alaska and eastern Siberia, thus over areas influenced by changes in zonal circulation, consistent with the results of the precipitation analyses. As in the case of precipitation, volcanic forcing produces global negative anomalies in the model ensemble that tend to last over 2-3 years.

     scPDSI shows a very similar EOF pattern to that of P-E, but with larger emphasis on the negative loadings that dominate over

larger scales, extending over Australia, the broader African continent, south America and large areas of North America and Eurasia (Fig. 15a). Thus, contrary to the P-E and precipitation results, which show more areas with positive anomalies during the 20th century, scPDSI shows for most regions negative trends. Negative scores in the PC during the volcanic episodes, associated to global spread negative loadings, are indicative of wetter soils, and indeed Fig. 15d shows increments in the global average of scPDSI in the model ensemble over timescales of about 5 years. Correlations between various ensemble members

are high for P-E (Fig. 14b) if the 20th-century trends are considered and lower and often not significant for pre-industrial times, indicating the influence of internal variability as in the case of precipitation (Fig. 10b). scPDSI correlations are however significant and high during pre-industrial times as well, likely a sign of the influence of temperature evolution in this variable. MCA-LIA changes for P-E (Fig. 14c) are very similar to the continental component of the corresponding precipitation pattern (Fig. 10c). As in the case of the EOF loadings, Fig. 15c shows MCA-LIA changes consistent with Fig. 14c, though with

widespread negative scPDSI values.

     The analysis of soil moisture (Fig. 16) does not show a clear relationship to external forcing, with some of the simulations in the ensemble showing poor correlations with temperature (Table 2). Some sensitivity to volcanic events and 20th-century warming is apparent in the behavior of the PC time series. The composite of volcanic eruptions shows some increase of variability in the ensemble but not a clear response to increasing or decreasing soil moisture during volcanic events, indicating

that internal variability may dominate over a potential response to the forcing. The associated EOF and MCA-LIA map (Fig. 16a,c) show similarities to those of Fig. 14a,c and 15a,c, with negative values over northern South America and parts of central and southern Africa. Over northern Africa, soil moisture shows positive loadings in the EOF and wetter values in the MCA-LIA differences. Interestingly, all EOF (MCA-LIA) maps in Fig. 14, Fig. 15 and Fig. 16 show negative (positive) values over eastern Asia and positive (negative) values over eastern North America. Despite an absence of a universal response of soil moisture

to external forcing, some of the simulations are clearer responses. This is the case of the CESM and GISS models. When an analysis is carried out independently for these subensembles, the resulting PC series (Fig. 17) show a clear correspondence with the temperature PC. Their corresponding EOFs show however considerable spatial differences.

     These results suggest that a more focused analysis would be required to address the behaviour of drought related variables and specifically, soil moisture, also considering more ad hoc techniques and homogeneous definitions of the soil moisture

content. Soil moisture, scPDSI and P-E measure different parts of the hydrological cycle. Additionally, soil moisture behavior





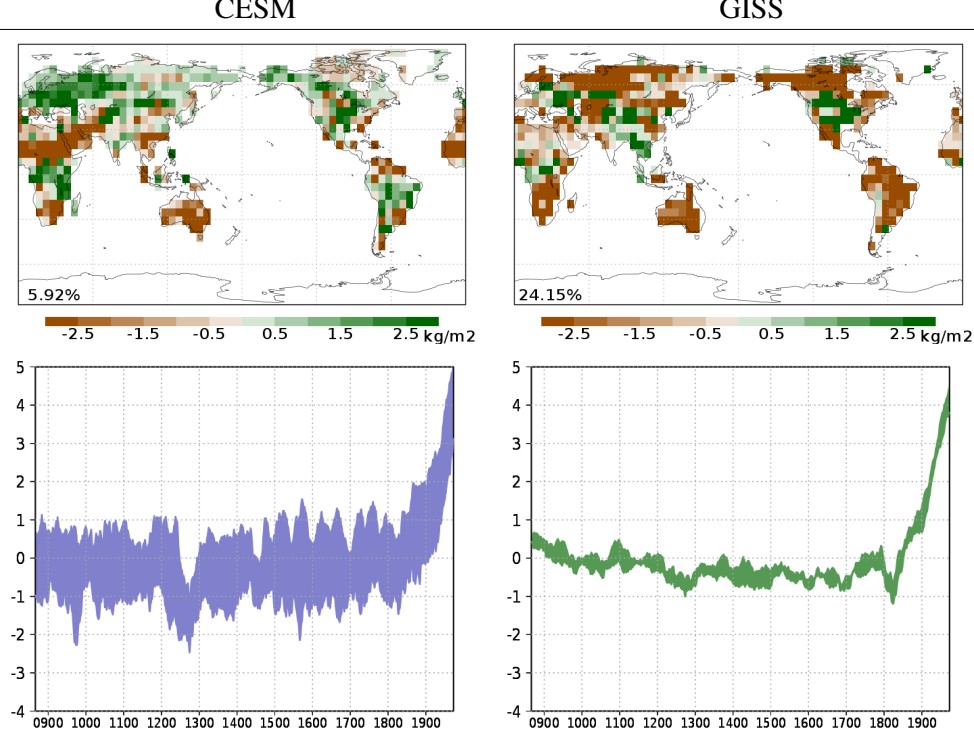

**Figure 17.** Second EOF and PC of soil moisture for the subensembles of CESM and GISS.

during the 20th century may be affected by $CO_2$ fertilization effects (Mankin et al., 2019), which are not present on other variables such as scPDSI. For this reason, the analyses based on these variables should be considered complementary and not necessarily comparable.

## 4   Conclusions

5   This work investigated the response of the current generation of climate models to changes in external forcing during the LM. For this purpose an ensemble of PMIP3/CMIP5 model simulations is considered, including all natural and anthropogenic forcings during the LM. It is focused on temperature as a reference to identify the response to external forcing and search for consistencies in the behaviour of large-scale dynamics and hydrology. Large-scale dynamics were assessed by studying changes in SLP and zonal and meridonal wind, while hydrological changes were studied by considering precipitation and

10   drought related variables. For the latter, P-E, scPDSI and soil moisture were considered. All variables were studied considering changes at different timescales. PC analysis was used to assess covariances between each variable and the temperature response and the external forcing signal at decadal to multicentennial timescales. MCA-LIA differences are characterized as descriptors



of large-scale changes associated with changes in natural forcing during pre-industrial times and volcanic composites are used as indicators of large interannual changes.

The temperature response to forcing depicts a spatial pattern of temperature anomalies that are larger over the continents and polar areas than over oceans (polar amplification). The temporal response of temperature shows changes that follow those of natural forcing during the LM, and anthropogenic forcing post 1850. All analysed variables, both related to dynamics and hydroclimate, show responses that correlate or are consistent with those of temperature. Changes in SLP depict increases (decreases) of the zonal flow in the high latitudes of both hemispheres during times of higher (lower) forcing. Within the tropical regions, zonal and meridional wind components indicate that changes favour convergence and alter the monsoon system. PC time series correlate with that of temperature, and volcanic composites also show sensitivity in these variables. The responses nevertheless can be spatially variable for different models and contribute differently to global averages.

Precipitation changes show a hydrological cycle that is enhanced, consistent with the changes in temperature. At mid and high latitudes, precipitation anomalies arise in response to changes in the zonal flow. Within the intertropical regions, precipitation anomalies distribute over monsoon regions and convergence zones, consistent with the changes described for climate change scenario simulations. Such large-scale anomalies distribute over different continents, generating covariance in intra and inter-basin regions. Nevertheless, other modes of internal variability also show widespread inter-basin anomalies and have the potential to contribute to multidecadal and centennial changes in the system.

The analysis of drought related variables shows dependencies with the definition of the variable itself and its relationship to temperature or precipitation. P-E shows responses that mimic those of precipitation over the continents. Increased P-E values tend to occur with increased forcing over mid and high-latitude regions influenced by the zonal flow. Within the intertropical regions the same anomalies for precipitation are simulated over monsoon sensitive areas. The pattern may change during different time intervals depending on the balance of precipitation and temperature effects as well as the effects of other modes of variability. For the MCA-LIA differences, increased drought is simulated over northern South America and southern Africa monsoon regions, while increased wetness is simulated over the Asian monsoons. These changes agree well with those in scPDSI MCA-LIA. The time evolution of its forcing response mode is very similar to that of P-E, but globally it produces increased scPDSI (reduced drought) during volcanic composites and contributes to increased drought during the 20th century.

The behaviour of soil moisture is more complex and model dependent. Some models depict clear responses to forcing but with differences in their spatial distribution and time response. The analysis of this variable shows a large dependency on the land models within the climate model itself. A response to external forcing is found when analysing subensembles of CESM and GISS, but not in the combined analyses including simulations from different models.

*Author contributions.* This study is part of PJRG's PhD. PJRG contributed with data processing, analysis of results and writing of the paper. JFGR, CMA and JES contributed to the analysis and discussion of results and to writing the paper.





*Competing interests.* The authors declare that they have no conflict of interest.

*Acknowledgements.* We gratefully acknowledge the IlModels (CGL2014-59644-R) and GreatModelS (RTI2018-102305-B-C21) projects.





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
