# Peer review of "Dynamical and hydrological changes in climate simulations of the last millennium"

_Climate of the Past, 2020_

## Referee Comment (RC1) · Anonymous Referee #1 · 30 Mar 2020

In this manuscript the authors analyse in a multi-model framework the typical responses of different key climate variables to the changes in the radiative forcing that occurred during the last millennium. This is nicely done by concatenating a large set of CMIP5/PMIP3 simulations, and computing the leading EOFs for several variables that describe different thermal, dynamical and hydrological aspects of the climate system. Since all concatenated simulations are driven with past estimates of the external radiative forcings, which synchronise some of the climate excursions across the different experiments, the EOFs extracted from the ensemble tend to successfully represent the common forced signal to all simulations. The analysis explores separately the long-term responses due to both anthropogenic and natural radiative forcing factors, as well

as the short-term impact of the largest volcanic eruptions.

I find the multi-model approach to be original and insightful, and the results of great interest. I thus recommend a minor revision of the manuscript, and enclose a list of comments that the authors would need to address to render the article suitable for publication in Climate of the Past.

General Comments:

1. The article is rather lengthy, and some parts feel repetitive. It would certainly benefit from some synthesis effort, so that the key messages are not obscured by the details. Some Figures could be removed, and their specific discussion in the main text shortened. For example, the most important changes in the hydrological cycle could be well described with just two variables: precipitation and the drought severity index. The P-E patterns are really close to those of precipitation (suggesting that precipitation is the dominant contributor to the surface freshwater fluxes over the continents). And soil moisture, as the authors already acknowledge in the paper, is not the most appropriate variable for inter-model comparison because different models compute it differently. And besides, it does not show a clear significant response to the forcings.

2. The global patterns of response (both the EOF and MCA-LIA composites) are beparticularly useful, as they help to easily identify the regions with the largest responses. But not so much the analyses based on zonal averages of dynamical and hydrological variables (Figures 8a and 11c), for which many of the regional features of the response are smoothed out. Indeed, it would be more interesting to address directly the response of the key indices that control these regional changes (ENSO/PDO, NAO, SAM,...). Plotting their associated time series, like in Fig 13, would allow to see how robust their forced signals across simulations are.

3. The volcanic impact analysis has also room for improvement. On one hand, it is currently focused on the global mean response, which makes sense for temperature (a variable that responds directly to changes in the radiative forcing), but not so much for

the dynamical and hydrological variables, whose response is, as I already mentioned, more regional. Focusing the plots on the regions with the largest response, as identified by the EOF or MCA-LIA composites, would help to identify stronger and more persistent influences of the volcanic eruptions to those in the global means. On the other hand, the current volcanic analysis is missing some estimate of statistical significance, which is essential to identify whether those responses are indeed meaningful. This could be done with a bootstrap approach that scans the periods with no volcanic eruptions to establish the significance threshold.

Specific comments:

- Page 2 Line 2: responses → changes

- Page 2 Line 5: consistently → consistent

- Page 3 Line 1: also have been → have also been

- Page 3 Line 10: the CMIP5/PMIP3

- Page 3 Lines 14-15: ", the Meteorological..., and with 13" → "and the Meteorological..., and 13l"

- Page 4 Figure 1 caption/Page 5 Line 10: composing → aggregating

- Page 5 Lines 28-30: The phrasing is confusing. I didn't really understand how it's done until I saw Figure 2d. The sentence suggests that compositing (or averaging) is not done with the five years before and 10 years after the volcanos, but it is instead done over the 12 main volcanic eruptions. And this is done for every year from the 5 preceding to the 10 following those volcanic eruptions.

- Page 5 Lines 31-33: Could you explain why is Gao's forcing used in some comes, and Crowley and Unterman's in others?

- Page 6 Table 2 Caption: temperature → surface temperature; of each → for each

- Page 6 Lines 1-3: Could you clarify if to make the multi-model concatenated array in which the EOF's are computed you first regrid all the experiments to a common grid? And to which one in that case? Otherwise the EOF array would be irregular in time. Or have the simulations been concatenated in space?

- Page 6 Line 4: Do you really apply an average? Or is it simply the EOF of the concatenated simulations? If there is no averaging it is better to refer to it as the "multi-model EOF".

- Page 7 Line 4: It would be more clear if you change "resulting EOFs" for "single experiment EOFs". Also, to prove that the differences are minor, you could compute the spatial pattern correlations between the individual EOFs and the multi-model one, and provide the range of correlation values in the text.

- Page 8 Figure 3 Caption: Also for clarity I would change "a PC analysis" → "the multi-model EOF analysis"

- Page 8 Lines 9-17: It feels weird to start your result section with a paragraph revising previous results. That's what the introduction is for. Previous results can also be discussed in the results section as well, but to contrast with your findings once they have been introduced. I strongly recommend to start directly with the second paragraph.

- Page 9 Lines 11-12: Not sure I agree. There are still important differences across members with the same model, which are hard to discern given the high line density in Figure 1c. To compare appropriately the forced vs internally driven temperature changes you would need, for a specific model ensemble, to compute the ensemble mean (which would describe the forced signal) and remove it from each of the individual members (to extract its internal variability component). I expect that many centennial changes will be of similar magnitude than the MCA-LIA transition. The exception should be the industrial warming trend, which will most probably remain unparalleled.

- Page 9 Lines 27-29: I suggest rephrasing the second sentence to make clear that

polar amplification is characteristic of the sea ice covered regions (via ocean/sea ice albedo feedbacks, among other processes) but not of the continental areas.

- Page 9 Lines 32-34: There is an important qualitative difference between Figure 2a,c that the authors do not comment. In the EOF, there is a stronger response in the Tropics than in the subtropics, that does not occur during the MCA-LIA transition. Could the authors discuss it, and the potential reasons?

- Page 10 Lines 7-8: Same as before. You start a subsection of results describing previous literature. Also, please note that the two modes of internal variability that you mention explicitly (ENSO and PDO) are coupled modes that involve the ocean, and therefore only partly related to atmospheric dynamics. It would make more sense to put forward the NAO, which is purely atmospheric and has been studied during the last millennium with different proxy reconstructions.

- Page 11 Lines 5-6: Could you specify what you mean by long term behaviour? The first PCs of SLP are basically characterised by a flat line and a positive trend starting in 1700. By contrast, the respective ones for surface temperature include strong multi-centennial oscillations, which for some models are of similar magnitude than the industrial warming trend.

- Page 11 Line 6-7: I wonder if the MCA-LIA difference that can be seen in Figure 4b is really significant. It does not seem to occur consistently for all the models. Indeed, another indication that the MCA-LIA difference is not a remarkable feature comes from the spread of correlations across model PCs in Figure 4b, which are only clearly above zero if the industrial era is considered.

- Page 11 Line 17: What do you mean by SLP stratification? Do you refer to the typical zonally-symmetric dipolar SLP response of SAM to global warming, with relative low surface pressure conditions at subpolar latitudes and high conditions at polar latitudes?

- Page 12 Line 3: There is not such a good similarity in the Southern Hemisphere.

Note for example that over Antarctica the response is of the opposite sign in the MCA-LIA pattern than in the EOF. age 12 Lines 4-5: The SAM intensifications/weakenings during the MCA/LIA are far from evident from Figure 4c. In particular, the significant response is not zonally-symmetric, and as mentioned before, Antarctica experiences a relative high during the MCA. If you really want to prove that MCA-LIA transition was accompanied by a weakening of the NAO/SAM, you should do show it with their respective indices.

- Page 13 Lines 32-34: There is no evident change from MCA to LIA in the PCs of the zonal wind. This implies that the EOF pattern mostly reflects the changes during the industrial period but not during the MCA.

- Page 13 Lines 34-35: To know if the EOF corresponds with a poleward displacement you need to show as well the mean zonal climatological winds. Otherwise, how can you tell that positive/negative loadings do not correspond to intensifications/weakenings of the climatological winds?

- Page 15 Lines 3-5: The spatial patterns in figure 7 show also important differences that should be acknowledged. For instance, in the North and Tropical Atlantic, or in the whole Pacific region.

- Page 18 Lines 9-10: Similar to the previous comment. In this case the response is really different in the Tropics.

- Page 18 Lines 16-17: I don't understand this statement. Figure 10 shows a positive response in North America, while climate projections suggest that the response is zero.

- Page 18 Line 17: Marine → Maritime

- Page 19 Line 2: You are not really showing consistency, just a multi-model response (which could be dominated by certain simulations/models)

- Page 20 Lines 1-2: As previously mentioned for the SLP patterns, shifts can only be diagnosed in relation to a climatological state, which has not been shown nor discussed.

- Page 20 Lines 4-5: The distribution is clearly centered at zero for all regions but EAS and SAS. For SAF there is a slight tendency to more positive values, but it could be happening by chance. A significance assessment would be helpful to draw more robust conclusions. You could, for instance, test if the median of the distribution is significantly different than zero.

- Page 20 Line 30: Strong statement. CCSM, HadCM , MRI and MPI don't really support this.

- Page 20 Line 35: There is no real agreement in the big picture in figure 12. Every model tends to have a different area of influence, which is particularly evident in the negative correlations.

- Page 24 Line 30: are → have

- Page 25 Lines 11-12: I find the phrasing of this sentence confusing. It's not clear if you refer to the covariability of all variables (including surface temperature) with the changes in the forcings or if you refer to the covariability between the PC related to the forcing of surface temperature, and the equivalent PCs for the other variables. I would simplify it just saying that "PC analysis was used to identify the multi-model typical pattern of response of different variables to the external forcing changes from decadal to multidecadal timescales"

- Page 26 Line 11: How can you tell that the hydrological is enhanced? Figures 14-17 simply show how the EOF of the forced modes of variability are, with regions of increased and regions of decreased precipitation.

---

## Referee Comment (RC2) · Pedro José Roldán-Gómez et al. · 17 Apr 2020

The paper by Roldan-Gomez and co-authors aims at evaluating the relative influence of external forcings on large-scale changes in PMIP2/CMIP5 last millennium climate model simulations including the historical period. To address this issue they relied on various statistical method and mainly EOFs analyzes and evolutions of their related PCs. Even though the paper is generally well written with potentially interesting results I have several concerns regarding the method and interpretations. The authors need to significantly improve the paper, as there are many important points to clarify or to be corrected before publication. I have listed bellow my main comments and criticism to be addressed:

[Figure]

Models and methods:

1. First of all they show time series covering the last millennium and the historical period as continuous model experiments. As far as I know this might not be the case for most of the model experiments used in this paper as the historical experiments in CMIP5 are branched off the pre-industrial control runs and are not a continuation of the LM simulations. The authors need to explain how they build the time series anomalies to make them look like seamless long climate model integrations. This is very important since this study discuss long-term trends and secular changes which depend on long term integration of external forcing histories. Historical runs branched of piControl runs might therefore include different initial mean background climate condition and trends. This should be clearly evaluated and the method used to take that into account when comparing to LM runs. How were the anomalies computed for each experiments used (piControl, LM, Historical) ?

2. The authors states that the model simulations were concatenated and time series low-pass filtered with a centered 31 years moving average. Which frequency cut-off was used to filter-out? The 31 years moving window was used to compute the anomalies? This should be clarified.

3. The method used to estimate the Total External Forcing (TEF) obtained by composing the contributions of several forcing factors should be explained in the method section.

4. This section does not give enough specific and explanations as how the EOFs analyzes is developed across PMIP3 models used. How the PC selection linked to the forcing is done? Which statistical method did you consider to evaluate the spurious results related to the different forcing data-sets and implementation strategies?

5. In the PMIP3 ensemble simulation, some model multiple realizations are included in the analyzes. From my understanding, each model experiments are given the same weight when performing the EOF analyzes or ensemble averaging. This will tend to

give mode weight to a few models. The authors state that the results are not affected by this sampling bias but they don't show and provide statistical measures in the subsequent analyzes to prove it. I suggest that a weighting is applied considering the number of experiments for each model to correct the sampling bias and make sure the results are unchanged.

6. The author state on page 8: "Some long-term changes in the external forcing, like the one during the transition from MCA to LIA, are significant enough to be obtained not only by performing PC analyzes but also by directly looking at the evolution of the variables during these two periods." I don't understand this sentence? Does that mean the authors assume that the leading PCs across LM ensemble for the considered variable and the actual evolution of the considered variable during the transition from MCA to LIA are the same? The authors should clarify this statement and prove it. Which long term external forcing changes during MCA/LIA are the authors referring to? This statement needs to be accompanied with quantified analyzes with statistical significance estimates.

Over the method section needs significant rewriting with a more systematic explanation of which methods is used to evaluated the statistical significance and relevance of the analyzes displayed in the results section. The authors should also clearly make a choice regarding the frequency window they want to investigate. Many mixed statements are presented in the results sections, regarding mean climate anomalies during the MCA relatively to LIA, secular trends and climate modes of variability occurring at various timescales. As it stands we cannot really makes sense and relate some assertion regarding climate modes of variability relying on displayed analyzes.

Results sections:

7. The authors make the following statement on page 8 in the 3.1 results section: "The peaks in volcanic forcing after the main eruptions are related to periods with lower global temperatures, while the multidecadal variability and long-term trends associated

with solar and anthropogenic forcings correspond with the long-term changes in temperatures that define periods of the MCA, LIA, and industrial era." Which analyzes attribute the multidecadal variability and long-term trends with solar and anthropogenic forcings? This is merely assertion not proven by presented results especially with latest forcing datasets used in PMIP3 which have shown a very weak or no fingerprint of solar irradiance forcing during the LM. The authors need to provide analyzes for the multidecadal variability and trends proving otherwise.

8. Page 9: "For the 20th century, all the analyzed simulations consistently show a warming, but trends strongly differ among simulations due to the different climate sensitivities of each model and the considered forcings". To which forcing this stronger sensitivity refers too? References should be cited to consolidate this assertion.

9. Page 9: "In a related and most relevant note, changes in the ensemble associated with external forcing are in general more relevant than those of internal variability." To which timescale this statement refers too? Is it for decadal or secular trends? This should be quantified and specifically quantified related to the frequency domain the authors want to discuss.

10. Page 9: "Note that most of the analyzed simulations show correlations larger than 0.5 and for simulations of the same model the correlations reach values around 0.9, both when analysing the whole period and when considering only the pre-industrial era. This indicates that even if the EOF has been obtained with a combined analysis, it is also representative of the individual simulations. Additionally, the use of large sets of simulations for some of the models, and 20 in particular the use of the 13 CESM-LME simulations, does not significantly bias the results, because the correlation ranges for models with individual simulations are as large as for the others." Since piControl runs are a measure of internal variability for each model, I don't understand why the authors get high correlation for both LM and piControl runs ? The method used should be clarified since the above results suggest either a flawed method or that LM changes and high correlations among model members including piControl are only due

to internal variability (the leading modes of internal variability present by construction in the piControl run?).

11. The authors also discuss changes in the leading EOF for SLP (and other hydro-climate variables) which probably reflects the first order thermodynamical response to global temperature changes due to external forcings. Yet the authors attribute it to changes in phases of the NAO, NAM and SAM or even ENSO/IPO in response to external forcings. They don't provide any analyzes that prove it. The authors states for example that there is "a tendency toward more positive phases of the NAO, NAM and SAM is observed during the MCA and industrial periods." However no relevant analyzes are shown to sustain these statements showing for example a quantified and causal link between the leading EOF for SLP and the actual changes in (internal) variability modes. The authors rather present long-term mean anomalies between MCA and LIA or time-series of leading PCs for global scale variables. Yet by definition internal modes of variability are characterized by leading pattern and frequencies prevalence that are not analyzes in the present paper. This comment applies almost to all the points discussed in the results section where many descriptive and speculative assertion.

12. For example, the presented and discussed results for SLP changes are confusing and somewhat contradictory. For instance, the authors state "simulations of GISS show an increase of pressure after volcanic events, while simulations of CESM-LME consistently show a decrease. This difference in the global average of pressure is not related to an opposite response in different models, but to the distribution of areas with positive and negative loadings in the mode of variability associated with the forcing. As shown in Fig. 5, simulations of CESM show a larger amount of areas with negative anomalies during periods with volcanic events, while simulations of GISS tend to show more areas with positive anomalies." An other example for the wind changes: "In spite of the differences in the global balance of regional positive and negative anomalies among models, all of them produce a global weakening in zonal circulation during

volcanic eruptions. " or "In general, this global analysis shows that regional modes of variability might be indirectly influenced by external forcing".

These are descriptive assertions, which need to be quantified and evaluated in terms of significance. Based on these few examples and the overall presentation of results sections, one can conclude that the simulation changes (leading EOF and volcanoes composites) are not really significant and alternatively interpreted as mean changes, decadal and secular trends or internal variability modes acting at inter annual (such as NAO) to decadal timescales (such as SAM) depending on the authors choice. Changes in variability modes are mixed with long-term trends and mean changes. However no results are presented and assessing these various questions separately depending on the timescale.

To sum-up I suggest major revisions. The authors need to exclude statements that are not sustained by actual relevant analyzes and focus only of long-term trends and mean MCA/LIA changes. In the actual form the paper will mislead the readers regarding the responses of the variability modes and the roles of external forcings based on speculative comments. The results presentations need to be improved focusing on specific timescale based on statistically significant signals analyzed with the appropriate method.

---

## Author Comment (AC1) · 13 May 2020

**Responses to reviewer's comments for "Dynamical and hydrological changes in climate simulations of the last millennium"**

We are grateful to the reviewers for their comments and suggestions, all of which have been helpful for improving the manuscript. We respond to each of the comments in our thorough replies below, providing in gray the comments from each review and in black our responses. Line and figure numbers correspond to the lines and figures in the newly revised manuscript unless otherwise noted.

**Reviewer 1:**

**R1C0**

In this manuscript the authors analyse in a multi-model framework the typical responses of different key climate variables to the changes in the radiative forcing that occurred during the last millennium. This is nicely done by concatenating a large set of CMIP5/PMIP3 simulations, and computing the leading EOFs for several variables that describe different thermal, dynamical and hydrological aspects of the climate system. Since all concatenated simulations are driven with past estimates of the external radiative forcings, which synchronise some of the climate excursions across the different experiments, the EOFs extracted from the ensemble tend to successfully represent the common forced signal to all simulations. The analysis explores separately the long-term responses due to both anthropogenic and natural radiative forcing factors, as well as the short-term impact of the largest volcanic eruptions. I find the multi-model approach to be original and insightful, and the results of great interest. I thus recommend a minor revision of the manuscript, and enclose a list of comments that the authors would need to address to render the article suitable for publication in Climate of the Past.

Following the comments from the reviewers, important changes have been done in the text to reduce its length (R1C1, R1C17 and R1C21), remove misleading statements (R1C36, R2C14, R2C15 and R2C16), and include the assessment of NAO, NAM and SAM indices (R1C2, R1C26 and R2C12), maps of climatology for SLP, zonal wind and precipitation (R1C26, R1C28 and R1C34), and the significance of SEA and the analyses of monsoon domains (R1C3 and R1C35).

General Comments:

**R1C1**

1. The article is rather lengthy, and some parts feel repetitive. It would certainly benefit from some synthesis effort, so that the key messages are not obscured by the details. Some Figures could be removed, and their specific discussion in the main text shortened. For example, the most important changes in the hydrological cycle could be well described with just two variables: precipitation and the drought severity index. The P-E patterns are really close to those of precipitation (suggesting that precipitation is the dominant contributor to the surface freshwater fluxes over the continents). And soil moisture, as the authors already acknowledge in the paper, is not the most appropriate variable for inter-model comparison because different models compute it differently. And besides, it does not show a clear significant response to the forcings.

Even if it is not the main contribution of the paper, the comparison of different variables representative of the hydroclimate, and in particular of the water content of the soil (P-E, scPDSI and soil moisture), is from our point of view an interesting result. Indeed, one conclusion from these analyses is that P-E is mostly affected by precipitation and PDSI by temperature, while soil moisture is very model-dependent. This is something that could be taken into account for future analyses based on the simulated hydroclimate. This follows an emerging convention to analyze a comprehensive suite of drought indicators, as done in other studies focused on drought projections

(Cook et al.,2020). We have tried to make this more clear in the new version, and hope the reviewer finds it convincing.

Regarding the length of the paper, we have removed the first paragraph of sections 3.1 and 3.2, according to comments R1C17 and R1C21. Following the same approach, we have removed those paragraphs with redundant information, including the sixth of section 3.2 and the first of section 3.3.

**R1C2**

2. The global patterns of response (both the EOF and MCA-LIA composites) are beparticularly useful, as they help to easily identify the regions with the largest responses. But not so much the analyses based on zonal averages of dynamical and hydrological variables (Figures 8a and 11c), for which many of the regional features of the response are smoothed out. Indeed, it would be more interesting to address directly the response of the key indices that control these regional changes (ENSO/PDO, NAO, SAM,...). Plotting their associated time series, like in Fig 13, would allow to see how robust their forced signals across simulations are.

To better represent the changes from MCA to LIA and its associated significance, several changes have been performed in the new version. Contours with climatology have been added to MCA-LIA maps according to comments R1C28 and R1C34, analyses of the latitudinal distribution of temperatures have been included in R1C20, significance of the changes in moonson domains have been added and commented according to R1C31 and R1C35, and changes in the text have been done in agreement with R1C23, R1C25 and R1C30. In this context, Fig. 9b and 12c show latitudinal changes that are useful to understand the impact of external forcing. The text associated with these figures has been changed to clarify its relation with the NAO, NAM and SAM indices: (P17 L14 - P18 L3) "*During the simulated MCA, changes in zonal and meridional winds took place with anti phase relationships that strengthened the zonal circulation at mid and high latitudes (Fig. 9b) with increases in the NAO, NAM and SAM (Fig 5), while within the intertropical regions the Trade winds and convergence were intensified (Fig. 7c and Fig. 9).*"

To support the discussions related to the NAO, NAM and SAM phenomena in Sect. 3.2, the associated indices have been computed and the percentage of positive phases for 50-year intervals has been included in Fig. 5. This figure shows a larger percentage of positive phases for the three indices during the MCA and industrial period, indicating significant changes in the zonal circulation. The text has been modified accordingly: (P15 L4-8) "*This pattern is associated with an intensification (MCA) and weakening (LIA) of the SAM and NAM/NAO, as shown in Fig. 5. The figure shows the percentage of years with positive NAO, NAM and SAM indices for successive intervals of 50 years. Consistent with the spatial patterns and temporal evolutions shown in the PC analysis, a tendency toward more positive phases of the NAO, NAM and SAM is observed during the MCA and industrial periods.*"

A description of how the NAO, NAM and SAM indices have been computed has been included in the methods section: (P10 L3-11) "*To better analyse the changes in the extratropical zonal circulation, the NAO, NAM and SAM indices have been computed. The NAO index has been obtained (Stephenson et al., 2006) with the difference of boreal winter (December, January and February; DJF) SLP average for (90°W to 60°E, 20°N to 55°N) and (90°W to 60°E, 55°N to 90°N), the NAM index was calculated (Li and Wang, 2003) as the difference between the DJF zonal mean SLP at 35°N and 65°N, and the SAM index was calculated from the difference between the zonal mean of annual SLP at 40°S and 65°S (Gong and Wang, 1999). The NAO, NAM and SAM indices have been obtained for each simulation in Table 1. The average of all the simulations was subsequently computed to determine the percentage of years with positive phases for successive intervals of 50 years. The change in the percentage of positive phases from the MCA to LIA was in*

*turn assessed and the significance of the changes evaluated using a student t-test.*"

**R1C3**

3. The volcanic impact analysis has also room for improvement. On one hand, it is currently focused on the global mean response, which makes sense for temperature (a variable that responds directly to changes in the radiative forcing), but not so much for the dynamical and hydrological variables, whose response is, as I already mentioned, more regional. Focusing the plots on the regions with the largest response, as identified by the EOF or MCA-LIA composites, would help to identify stronger and more persistent influences of the volcanic eruptions to those in the global means. On the other hand, the current volcanic analysis is missing some estimate of statistical significance, which is essential to identify whether those responses are indeed meaningful. This could be done with a bootstrap approach that scans the periods with no volcanic eruptions to establish the significance threshold.

The spatial distribution of SLP and zonal wind during volcanic events are respectively shown in Fig. 6 and Fig. 8. In these figures, it can be observed that for most regions the pattern resembles that of the EOF. The analysis of volcanic events is mainly focused on the global scale, but these analyses show that the impact of volcanic events is similar to that of the external forcing factors in the long term, and the spatial patterns obtained with the EOFs are therefore representative of the behavior during these events.

Regarding the significance of changes during volcanic events, we have added the significance level to Figures 2d, 4d, 7d, 11d, 15d, 16d and 17d, obtained with a bootstrap approach, by computing the percentile 5 and 95 of the distribution of averages generated with 2200 sets of 12 years (100 for each simulation) randomly selected from the whole period, excluding the years of volcanic eruptions and the ten years after them.

The description of this approach has been also included in the methods section: (P6 L3 - P7 L2) "*The significance of the changes in the variables evaluated within the SEA has been calculated using a bootstrap method. 2200 sets of 12 years (100 for each simulation) have been randomly taken from the whole analysed period, excluding the years of volcanic eruptions and the ten years after them, to generate a distribution of averages for each variable. The significance of the averages computed after the 12 volcanic eruptions are then determined using the 5 and 95 confidence limits from the bootstrap distribution.*"

Specific comments:

**R1C4**

- Page 2 Line 2: responses → changes

Changed.

**R1C5**

- Page 2 Line 5: consistently → consistent

Changed.

**R1C6**

- Page 3 Line 1: also have been → have also been

Changed.

**R1C7**
- Page 3 Line 10: the CMIP5/PMIP3

Changed to "*CMIP5/PMIP3*".

**R1C8**
- Page 3 Lines 14-15: ", the Meteorological. . ., and with 13" → "and the Meteorological. . ., and 13l"

Changed.

**R1C9**
- Page 4 Figure 1 caption/Page 5 Line 10: composing → aggregating

Changed.

**R1C10**
- Page 5 Lines 28-30: The phrasing is confusing. I didn't really understand how it's done until I saw Figure 2d. The sentence suggests that compositing (or averaging) is not done with the five years before and 10 years after the volcanos, but it is instead done over the 12 main volcanic eruptions. And this is done for every year from the 5 preceding to the 10 following those volcanic eruptions.

The paragraph has been rephrased to: (P5 L31-33) "*To assess the impact of such events on the climate, we use a Superposed Epoch Analysis (SEA), by defining a composite with the main volcanic eruptions within the LM and computing for this composite the global average of the variables previously mentioned for the five years before and ten years after the events*"

**R1C11**
- Page 5 Lines 31-33: Could you explain why is Gao's forcing used in some comes, and Crowley and Unterman's in others?

The idea is to use for the definition of the composite the dates of the largest eruptions that correspond to the actual forcing used for the simulation itself. The simulations of CESM-LME and CCSM were generated using Gao's forcing, so we considered more suitable to use this forcing also for the definition of the composite. In the other simulations, we selected Crowley and Unterman's forcing, as done in Masson-Delmotte et al. (2013).

We have made changes in the text to clarify this: (P5 L34 - P6 L1) "*For simulations of CESM-LME and CCSM, which use the reconstruction from Gao et al. (2008) as volcanic forcing, the years of the composite have been selected based on the minima of forcing from this reconstruction: 1452, 1584, 1600, 1641, 1673, 1693, 1719, 1762, 1815, 1883, 1963 and 1990.*"

**R1C12**
- Page 6 Table 2 Caption: temperature → surface temperature; of each → for each

Changed.

**R1C13**
- Page 6 Lines 1-3: Could you clarify if to make the multi-model concatenated array in which the EOF's are computed you first regrid all the experiments to a common grid? And to which one in that case? Otherwise the EOF array would be irregular in time. Or have the simulations been

concatenated in space?

Yes, the simulations have been regrided to a common grid before concatenating them. This procedure is explained in Sect. 2: (P5 L28-29) *"All simulations have been interpolated to a common 6ºx6º grid resolution, the coarsest among the analysed simulations."*

**R1C14**
- Page 6 Line 4: Do you really apply an average? Or is it simply the EOF of the concatenated simulations? If there is no averaging it is better to refer to it as the "multi-model EOF".

The multi-model EOF is not an average of the individual EOFs. We have removed the term "average EOFs", as it may be misleading.

We filter out the high frequency variability to focus on the low frequency response to external forcing by applying prior to the EOF analysis a 31-year running mean low pass filter, also as in previous work by the group (Fernandez-Donado et al 2013). Slight changes have been made to clarify this: (P7 L5-6) *"We concatenate all of the low-pass filtered simulations to determine the Empirical Orthogonal Functions (EOFs) across all of the models."*

**R1C15**
- Page 7 Line 4: It would be more clear if you change "resulting EOFs" for "single experiment EOFs" Also, to prove that the differences are minor, you could compute the spatial pattern correlations between the individual EOFs and the multi-model one, and provide the range of correlation values in the text.

Spatial correlations between single experiment EOFs and multi-model EOFs have been added: (P8 L5-7) *"The single experiment EOFs bear only regional differences that do not contradict the results obtained with the combined analyses, with spatial correlations with the multi-model EOFs reaching 0.9 for some simulations and 0.7 for most of them."*

**R1C16**
- Page 8 Figure 3 Caption: Also for clarity I would change "a PC analysis" → "the multi-model EOF analysis"

Changed.

**R1C17**
- Page 8 Lines 9-17: It feels weird to start your result section with a paragraph revising previous results. That's what the introduction is for. Previous results can also be discussed in the results section as well, but to contrast with your findings once they have been introduced. I strongly recommend to start directly with the second paragraph.

The paragraph has been removed. The definition of MCA and LIA according to Masson-Delmotte et al. (2013), which was included in that paragraph, has been moved to the previous section, since it is needed to understand the selection of years for the composites.

**R1C18**
- Page 9 Lines 11-12: Not sure I agree. There are still important differences across members with the same model, which are hard to discern given the high line density in Figure 1c. To compare appropriately the forced vs internally driven temperature changes you would need, for a specific model ensemble, to compute the ensemble mean (which would describe the forced signal) and remove it from each of the individual members (to extract its internal variability component). I

expect that many centennial changes will be of similar magnitude than the MCA-LIA transition. The exception should be the industrial warming trend, which will most probably remain unparalleled.

Thank you, this was a very useful suggestion. The internal variability component has been estimated by removing the ensemble average from each ensemble member. The results are included in Fig. R1.

[Figure]

**Figure R1.** Differences between each ensemble member simulation in Fig. 1c and the corresponding ensemble average. Dashed lines show $\bar{x} \pm 2s$ where $\bar{x}$ is the long-term mean of the residuals (zero) and $s$ the standard deviation.

Dashed lines in Fig. R1 show plus-minus the value of the residual standard deviation in the test suggested by the reviewer. The text has been modified accordingly: (P10 L22-31) "*These subensembles demonstrate that internal variability generates differences across simulations that are smaller than structural differences in model formulation across models. Figure 1c shows the range (dashed lines) of the residuals resulting from substracting the ensemble mean from each ensemble member simulation. Since the average of all ensemble members cancels out uncorrelated contributions of internal variability, the resulting ensemble mean constitues a smoothed estimation of the forced response and the residuals of substracting the ensemble mean from each ensemble member is an estimation of internal variability above 31-year timescales (Crowley, 2000; PAGES2k-PMIP3 group, 2015). Both the CESM and GISS ensembles in Fig. 1c show pre- and post-1850 low frequency changes larger than the estimated changes of internal variability. Changes in the ensemble associated with external forcing are therefore in general more relevant than those of internal variability above 31-year timescales.*"

**R1C19**
- Page 9 Lines 27-29: I suggest rephrasing the second sentence to make clear that polar amplification is characteristic of the sea ice covered regions (via ocean/sea ice albedo feedbacks, among other processes) but not of the continental areas.

The paragraph has been rephrased: (P11 L13-15) "*Regarding the spatial pattern of the EOF, values are larger over continental regions and smaller over oceans. For high latitudes, larger values are obtained over ice covered areas, consistent with the polar amplification response in climate change scenarios*"

**R1C20**
- Page 9 Lines 32-34: There is an important qualitative difference between Figure 2a,c that the authors do not comment. In the EOF, there is a stronger response in the Tropics than in the

subtropics, that does not occur during the MCA-LIA transition. Could the authors discuss it, and the potential reasons?

The difference is related to the different timescales of variability in tropical and extratropical areas. The tropical areas are more affected by high frequency variability, which is included in the EOF but not in the map of MCA-LIA differences that emphasize low frequency changes. This can be shown with the time series of temperature for different latitudes in Fig. R2. It can be observed that for extratropical areas the differences between the MCA and LIA are much larger than in the tropics.

**a) High Southern Latitude**

[Figure]

**b) Extratropical Southern Latitude**

[Figure]

**c) Tropical Latitude**

[Figure]

**d) Extratropical Northern Latitude**

[Figure]

**e) High Northern Latitude**

[Figure]

**Figure R2.** Time series of temperature for locations at different latitudes in the Pacific basin: **(a)** (-80º,180º), **(b)** (-40º,180º), **(c)** (0º,180º), **(d)** (40º,180º) and **(e)** (80º,180º).

The text has been modified to include this clarification: (P11 L19-22) "*The MCA-LIA pattern does*

*not emphasize the tropics as much as the EOF pattern, indicating that the low frequency variability changes in that area are minor and the higher tropical loadings in Fig. 2a stem from covariability at higher frequencies. Also note that area weighting has been applied for the EOF calculations, increasing the contribution of the tropical areas in these analyses.".*

**R1C21**

- Page 10 Lines 7-8: Same as before. You start a subsection of results describing previous literature. Also, please note that the two modes of internal variability that you mention explicitly (ENSO and PDO) are coupled modes that involve the ocean, and therefore only partly related to atmospheric dynamics. It would make more sense to put forward the NAO, which is purely atmospheric and has been studied during the last millennium with different proxy reconstructions.

The paragraph has been removed, since this information is already included in the introduction.

**R1C22**

- Page 11 Lines 5-6: Could you specify what you mean by long term behaviour? The first PCs of SLP are basically characterised by a flat line and a positive trend starting in 1700. By contrast, the respective ones for surface temperature include strong multi-centennial oscillations, which for some models are of similar magnitude than the industrial warming trend.

The explanation is included in the next sentence: (P11 L32-33) *"For the case of pressure, the average PC (black line in Fig. 4b) tends to show higher values during the MCA, lower during the LIA and a significant increase during the last century.".* The average PC in Fig. 4b shows larger values during the MCA than during the LIA. More details can be found in R1C27 and Table R1.

**R1C23**

- Page 11 Line 6-7: I wonder if the MCA-LIA difference that can be seen in Figure 4b is really significant. It does not seem to occur consistently for all the models. Indeed, another indication that the MCA-LIA difference is not a remarkable feature comes from the spread of correlations across model PCs in Figure 4b, which are only clearly above zero if the industrial era is considered.

The level of significance of the MCA-LIA is included in Fig. 4c. It shows significant changes for broad regions in northern latitudes (negative) and the tropics (positive). The significance of the PC correlations is also discussed in the text. Indeed MCA-LIA changes in SLP may be more subject to internal variability than temperature. Some changes have been made in the text to better represent the results: (P11 L31-33) *"The long-term behaviour of the first PC of pressure is comparable to that of the first PC of temperature. For the case of pressure, the average PC (black line in Fig. 4b) tends to show higher values during the MCA, lower during the LIA and a significant increase during the last century. "*

**R1C24**

- Page 11 Line 17: What do you mean by SLP stratification? Do you refer to the typical zonally-symmetric dipolar SLP response of SAM to global warming, with relative low surface pressure conditions at subpolar latitudes and high conditions at polar latitudes?

Yes, the text has been modified to make this clear.

**R1C25**

- Page 12 Line 3: There is not such a good similarity in the Southern Hemisphere. Note for example that over Antarctica the response is of the opposite sign in the MCA-LIA pattern than in the EOF.

Changed to: (P15 L1) *"As in the leading EOF, the spatial pattern of the MCA-LIA differences (Fig.*

*4c) also emphasizes the latitudinal gradients*".

**R1C26**
- Page 12 Lines 4-5: The SAM intensifications/weakenings during the MCA/LIA are far from evident from Figure 4c. In particular, the significant response is not zonally-symmetric, and as mentioned before, Antarctica experiences a relative high during the MCA. If you really want to prove that MCA-LIA transition was accompanied by a weakening of the NAO/SAM, you should do show it with their respective indices.

Climatological SLP has been added to Fig. 4c, according to R1C28. This shows that positive MCA-LIA differences appear over the maxima of SLP and negative differences over the minima, reinforcing the zonal circulation and contributing to more positive phases of the SAM.

As noted in R1C2, the NAO, NAM and SAM indices have been computed and the percentage of positive phases has been included in Fig. 5. For the three indices, a larger percentage of positive phases is obtained during the MCA and a smaller percentage during the LIA, indicating intensifications and weakenings, respectively.

The text has been modified accordingly: (P13 L5 - P14 L1) "*This spatial pattern, with positive loadings over the maxima of climatological SLP (black contours of Fig. 4a) and negative loadings over the minima (green contours of Fig. 4a), contributes to the positive phase of the mode to intensify gradients between subtropical and subpolar regions. This reinforces zonal circulation and contributes to more positive phases of the SAM (Jones et al., 2009; Fogt et al., 2009), as shown in Fig. 5.*", (P14 L2-5) "*Overall, higher positive (negative) loadings distribute over subtropical (polar) regions contributing to increase (decrease) the zonal flow during the MCA and industrial period, due to the slightly higher values of the PC, also consistent with NAM (Thompson and Wallace, 2001) enhancement.*"

**R1C27**
- Page 13 Lines 32-34: There is no evident change from MCA to LIA in the PCs of the zonal wind. This implies that the EOF pattern mostly reflects the changes during the industrial period but not during the MCA.

The changes from MCA to LIA in Fig.7b are evident in a decrease in the average PC (black line). Table R1 shows the difference in the average PC of temperature, SLP, zonal wind and precipitation between the MCA and LIA and the associated significance level ($p<0.05$). It can be seen that for all the variables the changes in the average PC from MCA to LIA are significant.

**Table R1.** Differences between MCA and LIA from the average PC time series of Fig. 2b, 4b, 7b and 11b. Significance level ($p<0.05$) is also included, obtained with a t-test for the difference of averages accounting for autocorrelation.

| Variable | MCA-LIA | Significance level (p<0.05) |
|---|---|---|
| Temperature | 1.12 | 0.33 |
| SLP | 0.24 | 0.16 |
| Zonal wind | 0.16 | 0.14 |
| Precipitation | 0.62 | 0.18 |

As discussed in R1C20, the MCA-LIA map emphasizes the low-frequency changes. During the MCA, subtropical westerlies are strengthened with respect to the LIA, consistent with the discussed

changes in the zonal circulation (see R1C2, R1C26, R1C28 and R2C12).

**R1C28**
- Page 13 Lines 34-35: To know if the EOF corresponds with a poleward displacement you need to show as well the mean zonal climatological winds. Otherwise, how can you tell that positive/negative loadings do not correspond to intensifications/weakenings of the climatological winds?

Contour lines for climatological SLP, zonal wind and precipitation have been added to Fig. 4, Fig. 7 and Fig.11. The positive (negative) climatological winds in Fig. 7 show the location of the Westerlies (Easterlies). It can be seen that positive anomalies are found in the high latitude side of the Westerlies, indicating a poleward displacement.

The text has been modified accordingly: (P16 L1-4) "*In the positive phase of the mode, negative (positive) loadings tend to distribute over the Easterlies (Westerlies) and over their high latitude side, thus increasing latitudinal gradients and contributing to a polar displacement of the wind system; trade winds are enhanced towards higher latitudes in the Atlantic and eastern Pacific.*"

The linkage between loadings and climatology also has been included for the case of SLP: (P13 L5 - P14 L1) "*This spatial pattern, with positive loadings over the maxima of climatological SLP (black contours of Fig. 4a) and negative loadings over the minima (green contours of Fig. 4a), contributes in the positive phase of the mode to intensify gradients between subtropical and subpolar regions. This reinforces zonal circulation and contributes to more positive phases of the SAM (Jones et al., 2009; Fogt et al., 2009), as shown in Fig. 5*"

**R1C29**
- Page 15 Lines 3-5: The spatial patterns in figure 7 show also important differences that should be acknowledged. For instance, in the North and Tropical Atlantic, or in the whole Pacific region.

That is the goal of these figures and the associated paragraph, to show that there exist differences between CESM and GISS subensembles in the spatial pattern during volcanic events. The paragraph is now rephrased to provide the details of the regions with differences: (P17 L8-11) "*In spite of the differences in some areas like the North and Tropical Atlantic and Pacific basins, both subensembles tend to weaken the global zonal circulation. However, the simulations of CESM (GISS) show more areas with positive (negative) zonal winds, which translate into a larger (smaller) increase of the global average.*"

**R1C30**
- Page 18 Lines 9-10: Similar to the previous comment. In this case the response is really different in the Tropics.

The text has been modified following this comment: (P20 L9-12) "*The MCA-LIA differences (Fig. 11c) show some similarities with the EOF loadings in extratropical regions, indicating larger precipitation at northern latitudes in the MCA (Fig. 11c) or with increased forcing at all timescales above 31 years (Fig. 11a,b). Within the tropical regions agreement is regionally complex, with MCA-LIA differences emphasizing low-frequency changes and EOF loadings including covariability at all timescales.*"

**R1C31**
- Page 18 Lines 16-17: I don't understand this statement. Figure 10 shows a positive response in North America, while climate projections suggest that the response is zero.

Figure 11 shows positive and negative responses over the NAMS. The text has been modified to make it clearer: (P20 L16 - P21 L2) *"MCA-LIA differences are regional in scope and show anomalies of different sign over the North and South American Monsoon Systems (NAMS, SAMS; Cerezo-Mota et al., 2011; Christensen et al., 2013), therefore without a clear response of NAMS and SAMS. This agrees with uncertainty in climate change projections over the NAMS, with CMIP5 models producing changes in precipitation that distribute around zero (Christensen et al., 2013). The same occurs over the Australian and Maritime Continent Monsoon Systems (AMSMC; Jourdain et al., 2013). Positive values are found over the East Asia and Southern Asian Summer Monsoon areas (EAS, SAS; May, 2011; Boo et al., 2011), in agreement with scenario simulations (Christensen et al., 2013)"*

**R1C32**
- Page 18 Line 17: Marine → Maritime

Changed.

**R1C33**
- Page 19 Line 2: You are not really showing consistency, just a multi-model response (which could be dominated by certain simulations/models)

In response to R1C15 and R2C5, we confirmed that the multi-model response is not biased to any simulation or model and could be considered representative of the response of the individual models. The correlations in Fig. 11b also show a good agreement among simulations, being many of the correlations significant even for the pre-industrial period. This indicates that the pattern of MCA-LIA shown in Fig. 11c is not dominated by a few simulations but common to many of them.

In any case, the sentence refers to the consistency of the spatial patterns with those obtained in scenario simulations (Christensen et al., 2013), and not to the consistency among different model simulations. The paragraph has been changed to clarify this: (P21 L2-4) *"Even if changes are not significant over many of these regions due to the large variability of precipitation, they show a pattern of response to forcing in LM PMIP3 simulations consistent with that of scenario simulations; consistency also extends to convergence zones."*

**R1C34**
- Page 20 Lines 1-2: As previously mentioned for the SLP patterns, shifts can only be diagnosed in relation to a climatological state, which has not been shown nor discussed.

Following comment R1C28, contours of climatological SLP, zonal wind and preciptiation have been added to Fig. 4, Fig. 7 and Fig.11. The climatological maxima of precipitation in Fig. 11 show that, over South America, the anomalies north of the maximum are negative and south of the maximum are positive. This indicates a shift of the rainfall, as discussed in the text.

The text has been modified to clarify this: (P21 L4-6) *"Over South America, negative anomalies are found in the northwest of the climatological maxima and positive anomalies in the southeast (Fig. 11a), depicting rainfall shifts in the South Atlantic Convergence Zone (Cavalcanti and Shimizu, 2012)."*

More details can be found in R1C2, R1C26 and R1C28.

**R1C35**
- Page 20 Lines 4-5: The distribution is clearly centered at zero for all regions but EAS and SAS. For SAF there is a slight tendency to more positive values, but it could be happening by chance. A

significance assessment would be helpful to draw more robust conclusions. You could, for instance, test if the median of the distribution is significantly different than zero.

Significance has been added in Fig. 12b. Text has been changed accordingly: (P22 L12-13) "*The largest impact of MCA-to-LIA transition in the monsoon systems appears over Asia, where EAS and SAS are significantly altered.*"

**R1C36**
- Page 20 Line 30: Strong statement. CCSM, HadCM , MRI and MPI don't really support this.

The statement has been changed to (P22 L35) "*Most model simulations correlate with external forcing over the same large-scale regions*"

**R1C37**
- Page 20 Line 35: There is no real agreement in the big picture in figure 12. Every model tends to have a different area of influence, which is particularly evident in the negative correlations.

The text has been modified as follows: (P23 L5 - P24 L1) "*Despite most of the models showing positive correlations in the extratropical and tropical areas of the Pacific basin, and negative correlations in tropical areas of the Atlantic basin and in Southeastern Asia, the areas of high correlation are spatially constrained to regional and even local scales and may not overlap in different models or even in simulations of the same model. These regional differences are likely the sign of the important influence of internal variability.*"

**R1C38**
- Page 24 Line 30: are → have

Changed.

**R1C39**
- Page 25 Lines 11-12: I find the phrasing of this sentence confusing. It's not clear if you refer to the covariability of all variables (including surface temperature) with the changes in the forcings or if you refer to the covariability between the PC related to the forcing of surface temperature, and the equivalent PCs for the other variables. I would simplify it just saying that "PC analysis was used to identify the multi-model typical pattern of response of different variables to the external forcing changes from decadal to multidecadal timescales"

Changed.

**R1C40**
- Page 26 Line 11: How can you tell that the hydrological is enhanced? Figures 14-17 simply show how the EOF of the forced modes of variability are, with regions of increased and regions of decreased precipitation.

The fact that the hydrological cycle is enhanced in situations of higher forcing can be observed in the EOF (Fig. 11a) and the map of MCA-LIA differences (Fig. 11c), where positive values are obtained mostly for monsoon and convergence areas where the climatological precipitation is larger (see also R1C34). Conversely, in analyses of volcanic eruptions (Fig. 11d), in which forcing is reduced, the global average of precipitation is also reduced.

This explanation is included in the text: (P22 L27-31) "*The responses described for different timescales in Fig. 11 are consistent with changes in scenario simulations described in Christensen*

*et al. (2013): increases in external forcing strengthen the hydrological cycle, enhancing zonal circulation in extratropical regions and increasing the global monsoon activity and equatorial convergence. This is found in the global average of precipitation after volcanic events and in the alteration of monsoons and latitudinal distribution obtained in the EOF, indicating a relevant response to external forcing in precipitation.”*

This behavior can be more clearly shown with the time series of average precipitation for the regions with more and less climatological precipitation, as included in Fig. R3. It can be observed that for the regions of lower precipitation (lower quartile of climatological precipitation; QL), no major differences are observed between situations of higher and lower forcing, while for the regions of larger precipitation (upper quartile of climatological precipitation; QU), important changes are observed, with generally larger values during the MCA than during the LIA and with less precipitation during volcanic events. This indicates that in situations of higher forcing the precipitation increases in the regions with more precipitation, showing therefore a hydrological cycle that is enhanced. This 'wet-get-wetter' effect is consistent with that of climate change projections (Christensen et al., 2013; Huang et al., 2013).

[Figure]

**Figure R3.** **(a)** Areas with climatological precipitation in the upper quartile (QU) and in the lower quartile (QL). Average of precipitation for **(b)** the whole globe, **(c)** the areas in the QU of climatological precipitation, and **(d)** the areas in the QL of climatological precipitation.

**Reviewer 2:**

**R2C0**

The paper by Roldan-Gomez and co-authors aims at evaluating the relative influence of external forcings on large-scale changes in PMIP2/CMIP5 last millennium climate model simulations including the historical period. To address this issue they relied on various statistical method and mainly EOFs analyzes and evolutions of their related PCs. Even though the paper is generally well written with potentially interesting results I have several concerns regarding the method and interpretations. The authors need to significantly improve the paper, as there are many important points to clarify or to be corrected before publication. I have listed bellow my main comments and criticism to be addressed:

We have included in the answers to the following comments clarifications and changes in the text and figures of the paper, mainly to complete the section of methods with a description of the exact experiments that have been considered for the analyses (R2C1), a more detailed description of the computation of TEF (R2C3), and a description of how the significance has been assessed at each timescale (R2C7), to remove descriptive paragraphs and statements (R1C1, R1C17, R1C21, R2C14, R2C15 and R2C16), and to include analyses based on NAO, NAM and SAM indices (R1C2, R1C26 and R2C12).

Models and methods:

**R2C1**

1. First of all they show time series covering the last millennium and the historical period as continuous model experiments. As far as I know this might not be the case for most of the model experiments used in this paper as the historical experiments in CMIP5 are branched off the pre-industrial control runs and are not a continuation of the LM simulations. The authors need to explain how they build the time series anomalies to make them look like seamless long climate model integrations. This is very important since this study discuss long-term trends and secular changes which depend on long term integration of external forcing histories. Historical runs branched of piControl runs might therefore include different initial mean background climate condition and trends. This should be clearly evaluated and the method used to take that into account when comparing to LM runs. How were the anomalies computed for each experiments used (piControl, LM, Historical) ?

To analyse the period from 850 to 2005 CE with CMIP5 simulations, the past1000 and historical experiments were concatenated, without performing any kind of post-processing. For some of the models (CSIRO, GISS and MPI) the historical simulations are derived from the past1000, but for others they are derived from the piControl, as indicated in the comment. This could generate a discontinuity in the input data in 1850 CE, when data from past1000 and historical experiment are concatenated. Figure R4 shows the global average of temperature, SLP and precipitation for the years between 1800 and 1900 CE, including the transition from past1000 to historical in 1850 CE. It can be observed that the discontinuity associated with the transition between experiments for these variables is not larger than the short-term variability of the data between 1800 and 1850 CE and between 1850 and 1900 CE. The impact of this transition is then removed when the short-term variability is filtered (by computing a 31-year moving average, as for the case of the analyses presented in the paper).

The text has been modified to include this information: (Table 1) "*All simulations span the period 850-2005 CE. This interval will be referred to herein as LM, even if within PMIP3 LM only includes 850-1850 CE. For the case of CMIP5/PMIP3 simulations, past1000 and historical experiments have been concatenated to cover this interval.*"

[Figure]

**Figure R4.** Global average of **(a)** unfiltered temperature, **(b)** 31-year low-pass filtered temperature, **(c)** unfiltered SLP, **(d)** 31-year low-pass filtered SLP, **(e)** unfiltered precipitation, and **(f)** 31-year low-pass filtered precipitation between 1800 and 1900 CE. The year 1850 CE, when input data changes from past1000 to historical experiments, is indicated with a vertical black line.

**R2C2**

2. The authors states that the model simulations were concatenated and time series low-pass filtered with a centered 31 years moving average. Which frequency cut-off was used to filter-out? The 31 years moving window was used to compute the anomalies? This should be clarified.

The only filter applied to the simulations is a 31 years moving average, which is applied to the input data before performing any analysis. This filter removes the variability in timescales shorter than 31 years. This approach has been used earlier to emphasize responses to external forcing changes

(Fernández-Donado et al., 2013; Luterbacher et al., 2016).

**R2C3**

More details about how the TEF is obtained have been added in the text: (P5 L9-14) "*The figure also shows an estimation of the Total External Forcing (TEF), obtained following Fernández-Donado et al. (2013) by aggregating the contributions of solar activity, orbital changes, volcanic activity, GHGs, including CO2 , CH4 and N2O, land use change, and anthropogenic sulfate aerosols, converted into radiative forcing units and filtered with a moving average of 31 years. Even if it presents some limitations in the conversion of volcanic forcing and the contribution of aerosols (Fernández-Donado et al., 2013), the TEF allows analyses of the long-term evolution of the overall incoming energy*"

A complete description is included and discussed in Fernández-Donado et al. (2013).

**R2C4**

To identify the PCs of each variable that are associated with the forcing we computed the correlation coefficients between all the PCs and the first PC of temperature, and selected those showing a larger correlation. The use of the first PC of temperature instead of the forcing time series removes the dependency on the particular reconstructions of forcing factors used by each model simulation. To ensure that the first mode of temperature is associated with the forcing, the correlations between the PCs of temperature and the time series of TEF for that model were also computed. These correlations and their significance are included in Table 2.

A paragraph has been added to explain this selection: (P9 L7-13) "*To identify which modes from those obtained in the PC analyses are capable of showing responses to external forcing, the correlation coefficients between their associated PC time series and the first PC of temperature have been computed, and only those showing the largest correlations have been analysed in detail. The use of the first mode of temperature instead of the time series of external forcing factors removes the dependency on the particular reconstructions used by each model simulation. To confirm that the first mode of temperature is linked to the external forcing for the analysed simulations, the correlation coefficient between the PC time series associated with this mode and the respective time series of TEF used for each specific model have been computed.*"

**R2C5**

Yes, in the analyses included in the paper each model experiment is considered with the same weight.

Thank you for the suggestion, it is a good way to check that the analyses presented in the paper are not biased to the CESM-LME for the fact of using more simulations of that model. We have performed the same analyses but using the same weight for each model, independently of the number of experiments. The EOFs and PCs obtained with this weighting approach are included in Fig. R5, and the correlations between these EOFs and the ones obtained by using the same weight for all the experiments (Fig. 2a, 4a, 7a and 11a of the paper) are included in Table R2.

The results are very similar for temperature and variables of atmospheric dynamics, with correlations larger than 0.85. The maps show more differences for the case of precipitation, especially for the tropical areas, but the correlations are still significant and these differences do not contradict any of the conclusions presented in the paper.

The text has been modified accordingly: (P8 L7 - P9 L2) "*The analyses have been also repeated weighting simulations so that each model would have the same influence, and the results are consistent (not shown).*"

**Table R2.** Correlations between the EOFs computed with the same weight for each model (Fig. R5) and the EOFs computed with the same weight for each simulation (Fig. 2a, 4a, 7a and 11a). Significant correlations (p<0.05) are shown in bold. Significance of correlation coefficients has been obtained with a t-test corrected for spatial autocorrelation following Dutilleul (1993).

| Variable | EOFs | Correlation |
|---|---|---|
| Temperature | Fig. R5 and Fig. 2a | **0.98** |
| SLP | Fig. R5 and Fig. 4a | **0.95** |
| Zonal wind | Fig. R5 and Fig. 7a | **0.85** |
| Precipitation | Fig. R5 and Fig. 11a | **0.47** |

[Figure]

**Figure R5.** EOF and PC time series for each simulation, as well as the average PC of all the simulations (black line), of **(a)** temperature, **(b)** SLP, **(c)** zonal wind, and **(d)** precipitation, obtained with the same weight for each model. The percentage of explained variance is shown within the EOF map.

**R2C6**

6. The author state on page 8: "Some long-term changes in the external forcing, like the one during the transition from MCA to LIA, are significant enough to be obtained not only by performing PC analyzes but also by directly looking at the evolution of the variables during these two periods." I don't understand this sentence? Does that mean the authors assume that the leading PCs across LM ensemble for the considered variable and the actual evolution of the considered variable during the transition from MCA to LIA are the same? The authors should clarify this statement and prove it. Which long term external forcing changes during MCA/LIA are the authors referring to? This

statement needs to be accompanied with quantified analyzes with statistical significance estimates.

The text has been modified for clarity as follows: (P9 L15-16) "*For a more detailed analysis of the long-term changes during the transition from the MCA to LIA, composites for the MCA and LIA have been defined from the ensemble average of each variable.*"

See also R1C20 for related comments.

**R2C7**

Over the method section needs significant rewriting with a more systematic explanation of which methods is used to evaluated the statistical significance and relevance of the analyzes displayed in the results section. The authors should also clearly make a choice regarding the frequency window they want to investigate. Many mixed statements are presented in the results sections, regarding mean climate anomalies during the MCA relatively to LIA, secular trends and climate modes of variability occurring at various timescales. As it stands we cannot really makes sense and relate some assertion regarding climate modes of variability relying on displayed analyzes.

The text has been modified to describe in detail how the significance of changes has been assessed:

- For the SEA: (P6 L3 - P7 L2) "*The significance of the changes in the variables evaluated within the SEA has been calculated using a bootstrap method. 2200 sets of 12 years (100 for each simulation) have been randomly taken from the whole analysed period, excluding the years of volcanic eruptions and the ten years after them, to generate a distribution of averages for each variable. The significance of the averages computed after the 12 volcanic eruptions are then determined using the 5 and 95 confidence limits from the bootstrap distribution.*" More details can be found in R1C3.

- For the correlations of PCs: (P9 L13-14) "*The significance of these correlations has been assessed with a t-test for the correlation coefficient, using an effective number of degrees of freedom that considers the window of the moving average applied to the input data.*"

- For the MCA-LIA differences: (P10 L1-2) "T*he significance of these MCA-LIA differences is assesed by performing a t-test for the difference of averages between the MCA and LIA for each grid point.*" More details can be found in R1C26, R1C28 and R1C34.

- For the percentage of positive phases of NAO, NAM and SAM: (P10 L9-11) "*The change in the percentage of positive phases from the MCA to LIA was in turn assessed and the significance of the changes evaluated using a student t-test.*" More details can be found in R1C2, R1C26 and R2C12.

Regarding the analysis of different timescales, the interannual analyses with the SEA, the multidecadal and centennial analyses with the PCs, and the multicentennial analyses with the MCA-LIA differences are considered complementary. They cover the response to external forcing at a wide range of timescales.

Results sections:

**R2C8**

7. The authors make the following statement on page 8 in the 3.1 results section: "The peaks in volcanic forcing after the main eruptions are related to periods with lower global temperatures, while the multidecadal variability and long-term trends associated with solar and anthropogenic

forcings correspond with the long-term changes in temperatures that define periods of the MCA, LIA, and industrial era." Which analyzes attribute the multidecadal variability and long-term trends with solar and anthropogenic forcings? This is merely assertion not proven by presented results especially with latest forcing datasets used in PMIP3 which have shown a very weak or no fingerprint of solar irradiance forcing during the LM. The authors need to provide analyzes for the multidecadal variability and trends proving otherwise.

The paragraph has been removed, following comment R1C17.

**R2C9**
8. Page 9: "For the 20th century, all the analyzed simulations consistently show a warming, but trends strongly differ among simulations due to the different climate sensitivities of each model and the considered forcings". To which forcing this stronger sensitivity refers too? References should be cited to consolidate this assertion.

References have been added: (P10 L18-19) "*due to the different climate sensitivities of each model (Vial et al., 2013) and the considered forcings (see discussion in Fernández-Donado et al.,2013)*"

**R2C10**
9. Page 9: "In a related and most relevant note, changes in the ensemble associated with external forcing are in general more relevant than those of internal variability." To which timescale this statement refers too? Is it for decadal or secular trends? This should be quantified and specifically quantified related to the frequency domain the authors want to discuss.

The timescale has been added, and Fig. 1c has been updated to quantify this (more details can be found in R1C18): (P10 L29-31) "C*hanges in the ensemble associated with external forcing are therefore in general more relevant than those of internal variability above 31-year timescales.*"

**R2C11**
10. Page 9: "Note that most of the analyzed simulations show correlations larger than 0.5 and for simulations of the same model the correlations reach values around 0.9, both when analysing the whole period and when considering only the pre-industrial era. This indicates that even if the EOF has been obtained with a combined analysis, it is also representative of the individual simulations. Additionally, the use of large sets of simulations for some of the models, and 20 in particular the use of the 13 CESM-LME simulations, does not significantly bias the results, because the correlation ranges for models with individual simulations are as large as for the others." Since piControl runs are a measure of internal variability for each model, I don't understand why the authors get high correlation for both LM and piControl runs ? The method used should be clarified since the above results suggest either a flawed method or that LM changes and high correlations among model members including piControl are only due to internal variability (the leading modes of internal variability present by construction in the piControl run?).

As clarified in R2C1, piControl runs have not been used. The simulations for CMIP5/PMIP3 models extend over the past1000 (850-1850 CE) and continue over the historical (1851-2005 CE) time intervals; many of them (e.g. CESM-LME, GISS) without disruption in 1850-51. The correlations of the pre-industrial period (Fig. 2b; PRE) are therefore based on past1000 runs, while the correlations of the whole interval (Fig. 2b; ALL) are based on past1000+historical. Since the time series are low-pass filtered by a 31-year moving average, the emphasis is put on the response to external forcing (see R2C1 and R2C2 for more details).

**R2C12**
11. The authors also discuss changes in the leading EOF for SLP (and other hydroclimate variables)

which probably reflects the first order thermodynamical response to global temperature changes due to external forcings. Yet the authors attribute it to changes in phases of the NAO, NAM and SAM or even ENSO/IPO in response to external forcings. They don't provide any analyzes that prove it. The authors states for example that there is "a tendency toward more positive phases of the NAO, NAM and SAM is observed during the MCA and industrial periods." However no relevant analyzes are shown to sustain these statements showing for example a quantified and causal link between the leading EOF for SLP and the actual changes in (internal) variability modes. The authors rather present long-term mean anomalies between MCA and LIA or time-series of leading PCs for global scale variables. Yet by definition internal modes of variability are characterized by leading pattern and frequencies prevalence that are not analyzes in the present paper. This comment applies almost to all the points discussed in the results section where many descriptive and speculative assertion.

The PC analysis covers all the timescales from decadal to multi-centennial. The fact that the spatial patterns of the NAO, NAM and SAM appear in the EOF indicates that the first mode of SLP obtained with the PC anlaysis is showing part of the variability associated with these modes, and the associated PC time series are representative of the evolution of these modes over time. To better show these patterns, contours of climatological SLP has been added to Fig. 4a, according to R1C26, R1C28 and R1C34.

To provide more evidence of this, we have included in Fig. 5 the percentage of years with positive NAO, NAM and SAM indices for 50-year intervals. The text has been modified accordingly: (P15 L5-8) "*The figure shows the percentage of years with positive NAO, NAM and SAM indices for successive intervals of 50 years. Consistent with the spatial patterns and temporal evolutions shown in the PC analysis, a tendency toward more positive phases of the NAO, NAM and SAM is observed during the MCA and industrial periods.*" More details can be found in R1C2.

**R2C13**
12. For example, the presented and discussed results for SLP changes are confusing and somewhat contradictory. For instance, the authors state "simulations of GISS show an increase of pressure after volcanic events, while simulations of CESM-LME consistently show a decrease. This difference in the global average of pressure is not related to an opposite response in different models, but to the distribution of areas with positive and negative loadings in the mode of variability associated with the forcing. As shown in Fig. 5, simulations of CESM show a larger amount of areas with negative anomalies during periods with volcanic events, while simulations of GISS tend to show more areas with positive anomalies."

Figure 6 shows that simulations of GISS and CESM-LME have differences in the response of SLP during volcanic events in the regional scale. This does not contradict the conclusions extracted from the SEA, and changes in SLP during volcanic events are significant for both subensembles, as shown in Fig. 5d (more details about the significance of the SEA can be found in R1C3). We expect that the overall behavior of different climate models is similar, but this does not exclude the possibility of having important differences in their spatial patterns.

**R2C14**
An other example for the wind changes: "In spite of the differences in the global balance of regional positive and negative anomalies among models, all of them produce a global weakening in zonal circulation during volcanic eruptions. "

Figure 8 shows that there exist differences in the regional distribution of positive and negative winds anomalies during volcanic events in GISS and CESM-LME simulations. As clarified in R2C13, it is possible to have differences in the spatial patterns obtained with different models for SLP and winds during volcanic events but still conclude that they are impacted by changes in the

forcing, as shown in Fig. 4d and 7d.

The sentence has been modified to clarify that it refers only to GISS and CESM-LME, since for the other models these maps were not presented: (P15 L19-21) "*In spite of the differences in the global balance of regional positive and negative anomalies between GISS and CESM-LME simulations, both produce a global weakening in zonal circulation during volcanic eruptions.*"

**R2C15**
or "In general, this global analysis shows that regional modes of variability might be indirectly influenced by external forcing".
These are descriptive assertions, which need to be quantified and evaluated in terms of significance.

The sentence has been removed.

**R2C16**
Based on these few examples and the overall presentation of results sections, one can conclude that the simulation changes (leading EOF and volcanoes composites) are not really significant and alternatively interpreted as mean changes, decadal and secular trends or internal variability modes acting at inter annual (such as NAO) to decadal timescales (such as SAM) depending on the authors choice. Changes in variability modes are mixed with long-term trends and mean changes. However no results are presented and assessing these various questions separately depending on the timescale.

Regarding the significance of the changes:

- The significance of MCA-LIA differences has been included in Figures 2c, 4c, 7c, 11c, 15c, 16c and 17c, showing that the changes during the transtion from MCA to LIA are significant in many areas for all the analysed variables.

- In response to R1C3, significance has been added to the SEA of volcanic events (Figures 2d, 4d, 7d, 11d, 15d, 16d and 17d). It is found that the global changes during these events are significant for most of the simulations and most of the analysed variables.

Regarding the separation of timescales:

- SEA, PC and MCA-LIA analyses for each variable are discussed in separate paragraphs, and in the methods section the scope of each analysis is described.

- In response to R2C12, the NAO, NAM and SAM indices have been computed and the evolution of the percentage of positive phases of these modes has been included in Fig. 5. This extends the conclusions obtained from the PC analysis of SLP to annual and decadal timescales.

**R2C17**
To sum-up I suggest major revisions. The authors need to exclude statements that are not sustained by actual relevant analyzes and focus only of long-term trends and mean MCA/LIA changes. In the actual form the paper will mislead the readers regarding the responses of the variability modes and the roles of external forcings based on speculative comments. The results presentations need to be improved focusing on specific timescale based on statistically significant signals analyzed with the appropriate method.

The text has been revised, following the previous comments, to keep only those statements directly

related to the figures and analyses presented in the paper. According to R1C1, R1C17, R1C21 and R2C15, descriptive paragraphs and sentences in the results section were removed.

As commented in R1C2 and R2C12, a new figure associated with NAO, NAM and SAM was added, to support the discussion related to these modes. The selected approach allows conclusions for different timescales (with SEA, PC and MCA-LIA analyses, and now with NAO, NAM and SAM indices), and not only for the long-term.

Regarding the methodology, methods section has been completed following R1C2, R2C1, R2C4 and R2C7 to include a more detailed description of the methods and the way the significance of the changes shown with these methods is obtained.

**References:**

Cook, B. I., Mankin, J. S., Marvel, K., Williams, A. P., Smerdon, J. E., and Anchukaitis, K. J.: Twenty-first Century Drought Projections in the CMIP6 Forcing Scenarios, Earth's Future, https://doi.org/10.1029/2019EF001461, 2020.

Dutilleul, P., Clifford, P., Richardson, S., and Hemon, D.: Modifying the t Test for Assessing the Correlation Between Two Spatial Processes, Biometrics, 49-1, 305-314, 1993.

Huang, P., Xie, S. P., Hu, K., Huang, G., and Huang, R.:  Patterns of the seasonal response of tropical rainfall to global warming, Nature Geoscience, 6, 357–361, 2013.

Luterbacher, J. et al.: European summer temperatures since Roman times, Environ. Res. Lett., 11, 024001, 2016.

---

## Author Comment (AC2) · 13 May 2020

Please see detailed responses in the attached supplement.

Please also note the supplement to this comment:
https://www.clim-past-discuss.net/cp-2020-8/cp-2020-8-AC2-supplement.pdf